# Learning Latent Graph Structures and their Uncertainty

## Abstract

Graph neural networks use relational information as an inductive bias to enhance prediction performance. Not rarely, task-relevant relations are unknown and graph structure learning approaches have been proposed to learn them from data. Given their latent nature, no graph observations are available to provide a direct training signal to the learnable relations. Therefore, graph topologies are typically learned on the prediction task alongside the other graph neural network parameters. In this paper, we demonstrate that minimizing point-prediction losses does not guarantee proper learning of the latent relational information and its associated uncertainty. Conversely, we prove that suitable loss functions on the stochastic model outputs simultaneously grant solving two tasks: (i) learning the unknown distribution of the latent graph and (ii) achieving optimal predictions of the model output. Finally, we propose a sampling-based method that solves this joint learning task. Empirical results validate our theoretical claims and demonstrate the effectiveness of the proposed approach.

## 1 Introduction

Relational information processing has provided breakthroughs in the analysis of rich and complex data coming from, e.g., social networks, natural language, and biology. This side information takes various forms, from structuring the data into clusters to defining causal relations and hierarchies, and enables machine learning models to condition their predictions on dependency-related observations. In this context, predictive models take the form $y = f_\psi(x, A)$, where the input-output relation $x \mapsto y$ – modeled by $f_\psi$ and its parameters in $\psi$ – is conditioned on the relational information encoded in variable $A$. Graph Neural Networks (GNNs) [Scarselli et al., 2008] are one example of models of this kind that rely on a graph structure represented as an adjacency matrix $A$ and have been demonstrated successful in a plethora of applications [Fout et al., 2017; Shlomi et al., 2020]. Throughout this paper, we focus on predictors where $A$ is an adjacency matrix, although the theoretical results we develop are valid for $A$ being any discrete latent random variable.

Indeed, relational information is needed to implement such a relational inductive bias and, in some cases, it is provided at the application design phase. However, more frequently, such topological information is not rich enough to address the problem at hand, and – not seldom – it is completely unavailable. Therefore, Graph Structure Learning (GSL) emerges as an approach to learn the graph topology [Kipf et al., 2018; Franceschi et al., 2019; Yu et al., 2021; Fatemi et al., 2021; Zhu et al., 2021; Cini et al., 2023] alongside the predictive model $f_\psi$. This entails formulating a joint learning process that learns the adjacency matrix $A$ – or a parameterization of it – along with the predictor's parameters $\psi$. This can be achieved by optimizing a loss function, e.g., a point prediction measure based on the square or the absolute prediction error.

Different sources of uncertainty affect the graph structure learning process, including epistemic uncertainty in the data and variability inherent in the data-generating process. Learning appropriate models of the data-generating process can provide valuable insights into the modeled environment with uncertainty quantification enhancing explainability and interpretability, ultimately enabling more informed decision-making. Examples of applications are found in the study of infection and information spreading, as well as biological systems [Gomez Rodriguez et al., 2013; Lokhov, 2016; Deleu et al., 2022]. It follows that a probabilistic framework is appropriate to accurately capture the uncertainty in the learned relations whenever randomness affects the graph topology. Probabilistic

approaches have been devised in recent years. For instance, research carried out in Franceschi et al. [2019]; Zhang et al. [2019]; Elinas et al. [2020]; Cini et al. [2023] propose methods that learn a parametric distribution $P_A^\theta$ over the latent graph structure $A$. However, none of them have studied whether these approaches were able to learn a *calibrated* latent distribution $P_A^\theta$, properly reflecting the uncertainty associated with the learned topology.

In this paper, we address the joint problem of learning a predictive model yielding optimal point-prediction performance of the output $y$ and, contextually, a calibrated distribution for the latent adjacency matrix $A$. In particular, the novel contributions can be summarized as:

1. We demonstrate that models trained to achieve optimal point predictions do *not* guarantee calibration of the adjacency matrix distribution [Section 4].

2. We provide theoretical conditions on the predictive model and loss function that guarantee both distribution calibration and optimal point-predictions [Section 5].

3. We propose a theoretically grounded sampling-based learning method to address the joint learning problem [Section 5].

4. We empirically validate the theoretical developments and claims presented in this paper and show that the proposed method is indeed able to solve the joint learning task [Section 6].

## 2 RELATED WORK

**Graph Structure Learning**   GSL is often employed end-to-end with a predictive model to better solve a downstream task. Examples include applications within graph deep learning methods for static [Jiang et al., 2019; Yu et al., 2021; Kazi et al., 2022] and temporal data [Wu et al., 2019; 2020; Cini et al., 2023; De Felice et al., 2024]; a recent review is provided by Zhu et al. [2021]. Some approaches from the literature model the latent graph structure as stochastic [Kipf et al., 2018; Franceschi et al., 2019; Elinas et al., 2020; Shang et al., 2021; Cini et al., 2023], mainly as a way to enforce sparsity of the adjacency matrix. To operate on discrete latent random variables, Franceschi et al. [2019] utilize straight-through gradient estimations, Cini et al. [2023] rely on score-based gradient estimators, while Niepert et al. [2021] design an implicit maximum likelihood estimation strategy. A different line of research is rooted in graph signal processing, where the graph is estimated from a constrained optimization problem and the smoothness assumption of the signals [Kalofolias, 2016; Dong et al., 2016; Mateos et al., 2019; Coutino et al., 2020; Pu et al., 2021]. A few works from the Bayesian literature have tackled the task of estimating uncertainties associated with graph edges. The model-based approaches by Lokhov [2016]; Gray et al. [2020] are two examples tackling relevant applications benefiting from uncertainty quantification. Within the deep learning literature, Zhang et al. [2019] propose a Bayesian Neural Network (BNN) modeling the random graph realizations. Differently, Wasserman & Mateos [2024] develop a BNN designed over graph signal processing principles. While some results on the output calibration are exhibited, to the best of our knowledge, no guarantee or evidence of calibration of the latent variable is provided, which we study in this paper instead.

**Calibration of the model's output**   Research on model calibration has primarily focused on obtaining accurate and consistent predictions of the statistical properties of the target (random) variables $y$, from which uncertainty estimates on the model's predictions are derived. For discrete outputs, such as in classification tasks, Guo et al. [2017] investigated the calibration of modern deep learning models and proposed temperature scaling as a solution. Other techniques in the same context include Histogram Binning [Zadrozny & Elkan, 2001], Cross Entropy loss with label smoothing [Müller et al., 2019], and Focal Loss [Mukhoti et al., 2020]. For continuous output distributions, Laves et al. [2020] proposed $\sigma$ scaling, while Kuleshov et al. [2018] developed a technique inspired by Platt scaling. More recently, conformal prediction techniques [Shafer & Vovk, 2008] have gained popularity for providing confidence intervals in predictions. We stress that within this paper, we are mainly concerned with latent variable calibration, rather than output calibration, although the two are related to each other.

**Deep latent variable models**   Latent variables are extensively used in deep generative modeling [Kingma & Welling, 2013; Rezende et al., 2014], both with continuous and discrete latent variables [Van Den Oord et al., 2017; Bartler et al., 2019]. In deep models, latent random variables often

lack direct physical meaning, with only the outputs being collected for training. Therefore, studies mainly focused on maximizing the likelihood of the observed outputs in the training set, rather than calibrating the latent distribution. A few works proposed regularization of the latent space to improve stability and accuracy [Xu & Durrett, 2018; Joo et al., 2020], facilitate smoother transitions in the output when the latent variable is slightly modified [Hadjeres et al., 2017], and apply other techniques aimed at enhancing data generation or improving model performance in general [Connor et al., 2021].

To the best of our knowledge, no prior work has studied the joint learning problem of calibrating the latent graph distribution while achieving optimal point predictions.

## 3 PROBLEM FORMULATION

Consider a set of $N$ interacting entities and the data-generating process

$$\begin{cases} A \sim P_A^* \\ y = f^*(x, A) \end{cases} \tag{1}$$

where $y \in \mathcal{Y}$ is the system output obtained from input $x \in \mathcal{X}$ through function $f^*$ and conditioned on a realization of the latent adjacency matrix $A \in \mathcal{A} \subseteq \{0,1\}^{N \times N}$ drawn from distribution $P_A^*$; input $x$ is assumed to be drawn from any distribution $P_x^*$ and superscript $*$ refers to unknown entities. Each entry of the adjacency matrix $A$ is a binary value encoding the existence of a pairwise relation between two nodes. In the sequel, $x \in \mathcal{X} \subseteq \mathbb{R}^{N \times d_{in}}$ and $y \in \mathcal{Y} \subseteq \mathbb{R}^{N \times d_{out}}$ are stacks of $N$ node-level feature vectors of dimension $d_{in}$ and $d_{out}$, respectively, representing continuous inputs and outputs.

Given a training dataset $\mathcal{D} = \{(x_i, y_i)\}_{i=1}^n$ of $n$ input-output observations from (1), we aim at learning a probabilistic predictive model

$$\begin{cases} A \sim P_A^\theta \\ \hat{y} = f_\psi(x, A) \end{cases} \tag{2}$$

from $\mathcal{D}$, while learning at the same time distribution $P_A^\theta$ approximating $P_A^*$. The two parameter vectors $\theta$ and $\psi$ are trained to approximate distinct entities in (1), namely the distribution $P_A^*$ and function $f^*$, respectively. We assume

**Assumption 3.1.** The family $\{P_A^\theta\}$ of probability distributions $P_A^\theta$ parametrized by $\theta$ and the family of predictive functions $\{f_\psi\}$ are expressive enough to contain the true latent distribution $P_A^*$ and function $f^*$, respectively.

Assumption 3.1 implies that $f^* \in \{f_\psi\}$ and $P_A^* \in \{P_A^\theta\}$ but does not request uniqueness of the parameters vectors $\psi^*$ and $\theta^*$ such that $f_{\psi^*} = f^*$ and $P_A^{\theta^*} = P_A^*$. Under such assumption the minimum function approximation error is null and we can focus on the theoretical conditions requested to guarantee successful learning, i.e., achieving both optimal point predictions and latent distribution calibration. In Section 6.2, we empirically show that the theoretical results can extend beyond this assumption in practice.

**Optimal point predictions**   Outputs $y$ and $\hat{y}$ of probabilistic model (1) and (2) are random variables following push-forward distributions[1] $P_{y|x}^*$ and $P_{y|x}^{\theta,\psi}$, respectively. A single point prediction $y_{PP} \in \mathcal{Y}$ can be obtained through an appropriate functional $T[\cdot]$ as

$$y_{PP} = y_{PP}(x, \theta, \psi) \equiv T\left[P_{y|x}^{\theta,\psi}\right]. \tag{3}$$

For example, $T$ can be the expected value or the value at a specific quantile. We then define an *optimal predictor* as one whose parameters $\theta$ and $\psi$ minimize the expected *point-prediction loss*

$$\mathcal{L}^{point}(\theta, \psi) = \mathbb{E}_{x \sim P_x^*}\left[\mathbb{E}_{y \sim P_{y|x}^*}\left[\ell\big(y, y_{PP}(x, \theta, \psi)\big)\right]\right] \tag{4}$$

between the system output $y$ and the point-prediction $y_{PP}$, as measured by of a loss function $\ell : \mathcal{Y} \times \mathcal{Y} \to \mathbb{R}_+$.

---

[1]The distribution of $y = f^*(x, A)$ originated from $P_A^*$ and of $\hat{y} = f_\psi(x, A)$ originated from $P_A^\theta$.

Statistical functional $T$ is coupled with the loss $\ell$ as the optimal functional $T$ to employ given a specific loss $\ell$ is often known [Berger, 1990; Gneiting, 2011], when $P_{y|x}^{\theta,\psi}$ approximates well $P_{y|x}^*$. For instance, if $\ell$ is the Mean Absolute Error (MAE) the associated functional $T$ is the median, if $\ell$ is the Mean Squared Error (MSE) the associated functional is the expected value.

**Latent distribution calibration**   Calibration of a parametrized distribution $P_A^\theta$ requires learning parameters $\theta$, so that $P_A^\theta$ aligns with true distribution $P_A^*$. Quantitatively, a dissimilarity measure $\Delta^{cal} : \mathcal{P}_A \times \mathcal{P}_A \to \mathbb{R}_+$, defined over a set $\mathcal{P}_A$ of distributions on $\mathcal{A}$, assesses how close two distributions are. The family of $f$-divergences [Rényi, 1961], such as the Kullback-Leibler divergence, and the integral probability metrics [Müller, 1997], such as the maximum mean discrepancy [Gretton et al., 2012] are examples of such dissimilarity measures. In this paper, we are interested in those discrepancies for which $\Delta^{cal}(P_1, P_2) = 0 \iff P_1 = P_2$ holds. It follows that the latent distribution $P_A^\theta$ is *calibrated* on $P_A^*$ if it minimizes the *latent distribution loss*

$$\mathcal{L}^{cal} = \mathbb{E}_{x \sim P_x^*} \left[ \Delta^{cal} \left( P_A^*, P_A^\theta \right) \right], \tag{5}$$

or simply $\mathcal{L}^{cal} = \Delta^{cal} \left( P_A^*, P_A^\theta \right)$, when $A$ and $x$ are independent.

The problem of designing a predictive model (2) that both yields optimal point predictions (i.e., minimizes $\mathcal{L}^{point}$ in (4)) and calibrates the latent distribution (i.e., minimizes $\mathcal{L}^{cal}$ in (5)) is nontrivial for two main reasons. At first, as the latent distribution $P_A^*$ is unknown (and no samples from it are available), we cannot directly estimate $\mathcal{L}^{cal}$. Second, as shown in Section 4, multiple sets of $\theta$ parameters may minimize $\mathcal{L}^{point}$ without minimizing $\mathcal{L}^{cal}$.

# 4   LIMITATIONS OF POINT-PREDICTION OPTIMIZATION

In this section, we demonstrate that the optimization of a point prediction loss (Equation (4)) does not generally grant calibration of the latent random variable $A$.

**Proposition 4.1.** *Consider Assumption 3.1. Loss function $\mathcal{L}^{point}(\theta, \psi)$ in (4) is minimized by all $\theta$ and $\psi$ s.t. $T\left[ P_{y|x}^{\theta,\psi} \right] = T\left[ P_{y|x}^* \right]$ almost surely on $x$ and, in particular,*

$$\mathcal{L}^{point}(\theta, \psi) \text{ is minimal} \quad \overset{\Longleftarrow}{\nRightarrow} \quad P_{y|x}^{\theta,\psi} = P_{y|x}^*.$$

The proof of the proposition is given in Appendix A.1; we provide a counterexample for which calibration is not granted even when the processing function $f_\psi$ is equal to $f^*$ in Appendix A.2.

Figure 1 empirically demonstrates that optimizing point prediction losses does not necessarily guarantee distribution calibration. In particular, we compute different losses between data generated with a ground truth system model (model (1) with optimal parameter $\theta^*$) and outputs produced with a different model (model (2), with varying $\theta$ parameters). In red, the MAE is used as the loss function $\ell$ in the point prediction loss $\mathcal{L}^{point}$ of (4). Since all $\theta \geq 0.725$ produce statistically equivalent losses, this simple experiment demonstrates the inefficacy of minimizing $\mathcal{L}^{point}$ for latent distribution calibration. In blue, we show the loss we propose in the next section, which presents a minimum in $\theta^*$. The

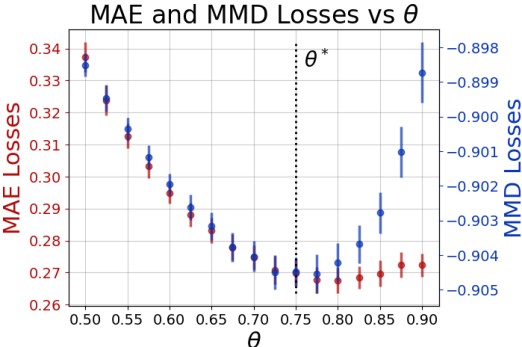

Figure 1: A data generating model, as in (1), is used to produce a dataset with latent distribution parameter $\theta^*$. Outputs are generated for different values $\theta$ as in (2). In red, losses are computed as in (4) with $\ell$ being the MAE. In blue, losses are computed with our approach described further on.

details of this experiment can be found in Section 6.1. However, we recommend reading the entire paper first to better understand the experiment's context and setting.

Given the provided negative result and the impossibility of assessing loss $\mathcal{L}^{cal}$ in (5), in the next section, we propose another optimization objective that, as we will prove, allows us to both calibrate the latent random variable and to have optimal point predictions.

## 5 PREDICTIVE DISTRIBUTION OPTIMIZATION: TWO BIRDS WITH ONE STONE

In this section, we show that we can achieve an optimal point predictor (2) and a calibrated latent distribution $P_A^\theta$ by comparing push-forward distributions $P_{y|x}^*$ and $P_{y|x}^{\theta,\psi}$ of the outputs $y$ conditioned on input $x$. In particular, Theorem 5.2 below proves that, under appropriate conditions, minimization of the *output distribution loss*

$$\mathcal{L}^{dist}(\theta,\psi) = \mathbb{E}_{x \sim P_x^*}\left[\Delta(P_{y|x}^*, P_{y|x}^{\theta,\psi})\right] \tag{6}$$

provides calibrated $P_A^\theta$, even when $P_A^*$ is not available; $\Delta : \mathcal{P}_y \times \mathcal{P}_y \to \mathbb{R}_+$ is a dissimilarity measure between distributions over space $\mathcal{Y}$. We assume the following on dissimilarity measure $\Delta$.

**Assumption 5.1.** $\Delta(P_1, P_2) \geq 0$ for all distributions $P_1$ and $P_2$ in $\mathcal{P}_y$ and $\Delta(P_1, P_2) = 0$ if and only if $P_1 = P_2$.

Several choices of $\Delta$ meet Assumption 5.1, e.g., $f$-divergences and some integral probability metrics [Müller, 1997]; the dissimilarity measure $\Delta$ employed in this paper is discussed in Section 5.1.

**Theorem 5.2.** *Let* $I = \{x : A \mapsto f^*(x, A) \text{ is injective}\} \subseteq \mathcal{X}$ *be the set of points* $x \in \mathcal{X}$ *such that map* $A \mapsto f^*(x, A)$ *is injective. Under Assumptions 3.1 and 5.1, if* $\mathbb{P}_{x \sim P_x^*}(I) > 0$, *then*

$$\mathcal{L}^{dist}(\theta,\psi^*) = 0 \implies \begin{cases} \mathcal{L}^{point}(\theta,\psi^*) \text{ is minimal} \\ \mathcal{L}^{cal}(\theta) = 0, \end{cases}$$

*where* $\psi^*$ *is such that* $f_{\psi^*} = f^*$.

Theorem 5.2 is proven in Appendix A.3. Under the theorem's hypotheses, a predictor that minimizes $\mathcal{L}^{dist}$ is both *calibrated* on the latent random distribution and provides *optimal point predictions*. This overcomes limits of Proposition 4.1 where optimization of $\mathcal{L}^{point}(\theta,\psi^*)$ does not grant $\mathcal{L}^{cal}(\theta) = 0$.

The hypotheses under which Theorem 5.2 holds are rather mild. In fact, condition $\mathbb{P}_{x \sim P_x^*}(I) > 0$ pertains to the data-generating process and intuitively ensures that, for some $x$, different latent random variables produce different outputs. A sufficient condition for $\mathbb{P}_{x \sim P_x^*}(I) > 0$ to hold is the existence of a point $\bar{x}$ in the support of $P_x^*$ such that $A \mapsto f^*(\bar{x}, A)$ is injective with $f^*$ continuous w.r.t. $\bar{x}$; see Corollary A.1 in Appendix A.3. Although only a single point $\bar{x}$ is required, having more points that satisfy the condition simplifies the training of the parameters. Corollary A.1 holds for arbitrarily complex processing functions $f^*$. More specifically, when considering simple GNN layers and discrete latent matrices $A$, we can prove that the condition $\mathbb{P}_{x \sim P_x^*}(I) > 0$ is − except from pathological cases − always satisfied (see Proposition A.2 in Appendix A.3). Instead, condition $f_\psi = f^*$ is set to avoid scenarios of different, yet equivalent,[2] representations of the latent distribution. An empirical analysis of the theorem's assumptions is provided in Section 6.2, demonstrating that the theoretical results hold in practice, even when the assumption does not strictly apply.

Assumptions 3.1 and 5.1 can be met with an appropriate choice of model (2) and measure $\Delta$; as such they are controllable by the designer. Assumption 5.1 prevents from obtaining mismatched output distributions when $\mathcal{L}^{dist}(\theta,\psi) = 0$ and can be easily satisfied. As mentioned above, popular measures, e.g., the Kullback-Leibler divergence, meet the theorem's assumptions and therefore can be adopted as $\Delta$. However, as $f$-divergences rely on the explicit evaluation of the likelihood of $y$, they are not always practical to compute [Mohamed & Lakshminarayanan, 2016]. For this reason, we propose considering the Maximum Mean Discrepancy (MMD) [Gretton et al., 2012] as a versatile alternative that allows Monte Carlo computation without requiring evaluations of the likelihood w.r.t. the output distributions $P_{y|x}^*$ and $P_{y|x}^{\theta,\psi}$. Energy distances [Székely & Rizzo, 2013] provide an alternative feasible choice.

### 5.1 MAXIMUM MEAN DISCREPANCY

Given two distributions $P_1, P_2 \in \mathcal{P}_y$, MMD can be defined as

$$\mathrm{MMD}_{\mathcal{G}}[P_1, P_2] = \sup_{g \in \mathcal{G}}\left\{\mathbb{E}_{y \sim P_1}[g(y)] - \mathbb{E}_{y \sim P_2}[g(y)]\right\}, \tag{7}$$

---

[2]E.g., $f_\psi(A, x) = f_*(\mathbf{1} - A, x)$ and $P_A^\theta$ encoding the absence of edges instead of their presence as in $P_A^*$.

i.e., the supremum, taken over a set $\mathcal{G}$ of functions $\mathcal{Y} \to \mathbb{R}$, of the difference between expected values w.r.t. $P_1$ and $P_2$. An equivalent form is derived for a generic kernel function $\kappa(\cdot, \cdot) : \mathcal{Y} \times \mathcal{Y} \to \mathbb{R}$:

$$\text{MMD}_{\mathcal{G}_\kappa}^2[P_1, P_2] = \mathop{\mathbb{E}}_{y_1, y_1' \sim P_1}\Big[\kappa(y_1, y_1')\Big] - 2\mathop{\mathbb{E}}_{\substack{y_1 \sim P_1 \\ y_2 \sim P_2}}\Big[\kappa(y_1, y_2)\Big] + \mathop{\mathbb{E}}_{y_2, y_2' \sim P_2}\Big[\kappa(y_2, y_2')\Big] \quad (8)$$

and it is associated with the unit-ball $\mathcal{G}_k$ of functions in the reproducing kernel Hilbert space of $\kappa$; note that (8) is the square of (7). Moreover, when universal kernels are considered (e.g., the Gaussian one), then (8) fulfills Assumption 5.1 (see Theorem 5 of Gretton et al. [2012]). Dissimilarity in (8) can be conveniently estimated via Monte Carlo (MC) and employed within a deep learning framework. Accordingly, we set $\Delta = \text{MMD}_{\mathcal{G}_\kappa}^2$ and learn parameter vectors $\psi$ and $\theta$ by minimizing $\mathcal{L}^{dist}(\theta, \psi)$ via gradient-descent methods.

## 5.2 Finite-sample computation of the loss

To compute the gradient of $\mathcal{L}^{dist}(\theta, \psi) = \mathbb{E}_{x \sim P_x^*}\Big[\text{MMD}_{\mathcal{G}_\kappa}^2\Big[P_{y|x}^{\theta, \psi}, P_{y|x}^*\Big]\Big]$ w.r.t. parameter vectors $\psi$ and $\theta$, we rely on MC sampling to estimate in (6) expectations over input $x \sim P_x^*$, target output $y \sim P_{y|x}^*$ and model output $\hat{y} \sim P_{y|x}^{\theta, \psi}$. This amounts to substitute $\text{MMD}_{\mathcal{G}_\kappa}^2$ with

$$\widehat{\text{MMD}}^2(\theta, \psi; x, y) = \frac{\sum_{i=1}^{N_{adj}} \sum_{j=1}^{i-1} \kappa(\hat{y}_i, \hat{y}_j)}{N_{adj}(N_{adj} - 1)} - 2\frac{\sum_{i=1}^{N_{adj}} \kappa(y, \hat{y}_i)}{N_{adj}} \quad (9)$$

In (9), $N_{adj} > 1$ is the number of adjacency matrices sampled from $P_A^\theta$ to obtain output samples $\hat{y}_i = f_\psi(x, A_i) \sim P_{y|x}^{\theta, \psi}$, whereas the pair $(x, y)$ is a pair from the training set $\mathcal{D}$. We remark that in (9) the third term of (8) – i.e., the one associated with the double expectation with respect to $P_{y|x}^*$ – is neglected as it does not depend on $\psi$ and $\theta$.

Gradient $\nabla_\psi \mathcal{L}^{dist}(\theta, \psi)$ is computed via automatic differentiation by averaging $\nabla_\psi \widehat{\text{MMD}}^2(\theta, \psi)$ within a mini-batch of observed data pairs $(x_i, y_i) \in \mathcal{D}$. For $\nabla_\theta \mathcal{L}^{dist}(\theta, \psi)$, the same approach is not feasible. This limitation arises because the gradient is computed with respect to the same parameter vector $\theta$ that defines the integrated distribution. Here, we rely on a score-function gradient estimator (SFE) [Williams, 1992; Mohamed et al., 2020] which uses the log derivative trick to rewrite the gradient of an expected loss $L$ as $\nabla_\theta \mathbb{E}_{A \sim P^\theta}[L(A)] = \mathbb{E}_{A \sim P^\theta}[L(A)\nabla_\theta \log P^\theta(A)]$, with $P^\theta(A)$ denoting the likelihood of $A \sim P^\theta$. Applying the SFE to our problem the gradient of the loss function w.r.t. $\theta$ reads:

$$\nabla_\theta \mathcal{L}^{dist}(\psi, \theta) = \mathop{\mathbb{E}}_{(x,y^*) \sim P_{x,y}^*}\left[\mathop{\mathbb{E}}_{\hat{y}_1, \hat{y}_2 \sim P_{y|x}^{\theta, \psi}}\Big[\kappa(\hat{y}_1, \hat{y}_2)\nabla_\theta \log\Big(P_{y|x}^{\theta, \psi}(\hat{y}_1)P_{y|x}^{\theta, \psi}(\hat{y}_2)\Big)\Big]\right.$$
$$\left. - 2\mathop{\mathbb{E}}_{\hat{y} \sim P_{y|x}^{\theta, \psi}}\Big[\kappa(y^*, \hat{y})\nabla_\theta \log P_{y|x}^{\theta, \psi}(\hat{y})\Big]\right] \quad (10)$$

An apparent setback of SFEs is their high variance [Mohamed et al., 2020], which we address in Section 5.3 by deriving a variance-reduction technique based on control variates that requires negligible computational overhead.

## 5.3 Variance-reduced loss for SFE

Two natural approaches to reduce the variance of MC estimates of (10) involve (i) increasing the number $B$ of training data points in the mini-batch used for each gradient estimate and (ii) increasing the number $N_{adj}$ of adjacency matrices sampled for each data point in (9). These techniques act on two different sources of noise. Increasing $B$ decreases the variance coming from the data-generating process, whereas increasing $N_{adj}$ improves the approximation of the predictive distribution $P_{y|x}^{\theta, \psi}$. Nonetheless, by fixing $B$ and $N_{adj}$, it is possible to further reduce the latter source of variance by employing the *control variates* method [Mohamed et al., 2020] that, in our case, requires only a negligible computational overhead but sensibly improves the training speed (see Section 6).

Consider the expectation $\mathbb{E}_{A \sim P^\theta}[L(A)\nabla_\theta \log P^\theta(A)]$ of the SFE – both terms in (10) can be cast into that form. With the control variates method, $L(A)$ is replaced by a surrogate function

$$\tilde{L}(A) = L(A) - \beta\Big(h(A) - \mathbb{E}_{A \sim P^\theta}[h(A)]\Big) \tag{11}$$

that leads to a reduced variance in the MC estimator while maintaining it unbiased. In this paper, we set function $h(A)$ to $\nabla_\theta \log P^\theta(A)$ and show how to compute a near-optimal choice for scalar value $\beta$, often called *baseline* in the literature. As the expected value of $\nabla_\theta \log P^\theta(A)$ is zero, gradient (10) rewrites as

$$\nabla_\theta \mathcal{L}^{dist} = \mathop{\mathbb{E}}_{(x,y^*) \sim P_{x,y}^*}\left[ \mathop{\mathbb{E}}_{A_1,A_2 \sim P_A^\theta} \Big[ (\kappa(f_\psi(x,A_1), f_\psi(x,A_2)) - \beta_1)\ \nabla_\theta \log\big(P_A^\theta(A_1)P_A^\theta(A_2)\big) \Big] \right.$$
$$\left. - 2 \mathop{\mathbb{E}}_{A \sim P_A^\theta} \Big[ (\kappa(y^*, f_\psi(x,A)) - \beta_2)\ \nabla_\theta \log P_A^\theta(A) \Big] \right]. \tag{12}$$

In Appendix B, we show that in our setup the best values of $\beta_1$ and $\beta_2$ are approximated by

$$\tilde{\beta}_1 = \mathop{\mathbb{E}}_{\substack{x \sim P_x^* \\ A_1,A_2 \sim P_A^\theta}} \Big[ \kappa\big(f_\psi(x,A_1), f_\psi(x,A_2)\big) \Big], \qquad \tilde{\beta}_2 = \mathop{\mathbb{E}}_{\substack{(x,y^*) \sim P_{x,y}^* \\ A \sim P_A^\theta}} \Big[ \kappa\big(y^*, f_\psi(x,A)\big) \Big], \tag{13}$$

which can be efficiently computed via MC reusing the kernel values already computed for (12).

## 5.4 COMPUTATIONAL COMPLEXITY

Focusing on the most significant terms, for every data pair $(x, y)$ in the training set, computing the loss $\mathcal{L}^{dist}$ requires $\mathcal{O}(N_{adj}^2)$ kernel evaluations $\kappa(\hat{y}_i, \hat{y}_j)$ in (9), $\mathcal{O}(N_{adj})$ forward passes through the GNN $\hat{y}_i = f_\psi(x, A_i)$ in (9) and $\mathcal{O}(N_{adj})$ likelihood computations $P_A^\theta(A_i)$ in (12). The computation of baselines $\beta_1$ and $\beta_2$ in (13) requires virtually no overhead, as commented in previous Section 5.3. Similarly, computing the loss gradients requires $\mathcal{O}(N_{adj}^2)$ derivatives for what concerns the kernels, $\mathcal{O}(N_{adj})$ gradients $\nabla_\psi \hat{y}_i$ and $\nabla_\theta \log P_A^\theta(A_i)$. We empirically observed that for $N_{adj} \geq 16$, both the latent distribution loss $\mathcal{L}^{cal}$ and the point prediction loss $\mathcal{L}^{point}$ of final models are equivalent for the considered problem. This suggests that $N_{adj}$ is not a critical hyperparameter.

Since we can employ sparse representations of adjacency matrices, the GNN processing costs scale linearly in the number of nodes $N$ for bounded-degree graphs. From our experience, the GNN processing is the most demanding operation and the cost of quadratic components, such as the parameterization of $\theta_{ij}$, do not pose significant overhead.

## 6 EXPERIMENTS

This section empirically validates the proposed technique and the main claims of the paper. To assess the calibration performance of models, it is necessary to compare the learned graph distribution $P_A^\theta$ with the ground-truth latent distribution $P_A^*$. However, to our knowledge, no real-world datasets provide such ground truth. Therefore, we developed theoretical guarantees to support the application of these methods to real data and – in this section – we conduct the empirical validation using synthetic data. Section 6.1 demonstrates that the proposed approach can successfully solve the joint learning problem across different graph sizes, highlights the effectiveness of the variance reduction technique, and reveals challenges in optimizing point prediction losses when also aiming for latent variable calibration. Section 6.2 empirically investigates the generality of the theoretical results we develop, demonstrating appropriate calibration of the latent distribution even in scenarios where the assumptions of Theorem 5.2 do not hold.

In order to assess the latent variable calibration performance, i.e., the discrepancy between $P_A^*$ and the learned $P_A^\theta$, the ground-truth latent distribution $P_A^*$ must be given. Such ground-truth knowledge is not available in real-world applications as the latent distribution is indeed unknown. For this reason, we designed a synthetic dataset that allows us to evaluate different performance metrics on both $y$ and $A$ while controlling several properties of the task, like the number of nodes and the probability of each edge. We remark that the latent distribution $P_A^*$ is used *only* to assess performance and does not drive the model training in any way.

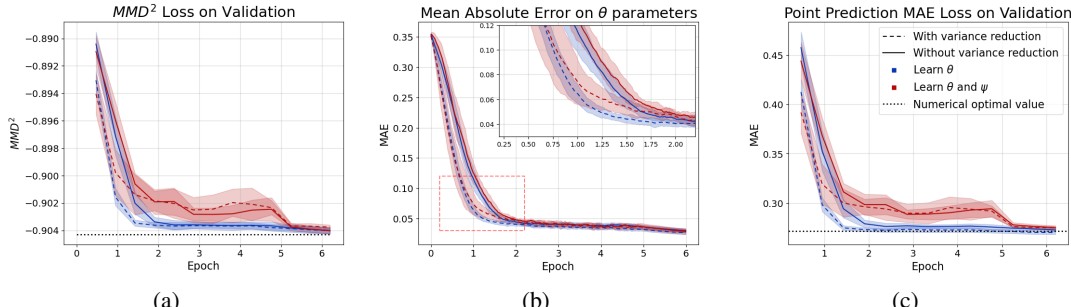

(a)                                  (b)                                  (c)

Figure 2: Validation losses $\mathcal{L}^{dist}$, $\mathcal{L}^{cal}$ and $\mathcal{L}^{point}$ during training. At epoch 5, the learning rate is decreased to ensure convergence. $\mathcal{L}^{dist}$ in Subfigure 2a is negative as the third term in (8) is constant and not considered.

**Dataset and models**  Consider data-generating process (1) with latent distribution $P_A^* = P_A^{\theta^*}$ producing $N$-node adjacency matrices. $P_A^*$ is defined by a set of $N \times N$ independent Bernoulli distributions, each of which corresponds to the sampling probability of an edge. Function $f_* = f_{\psi^*}$ is a generic GNN with node-level readout, i.e., $f_{\psi^*}(\cdot, A) : \mathbb{R}^{N \times d_{in}} \to \mathbb{R}^{N \times d_{out}}$. In the below experiments, $N$ is set to 12, while input and output node feature dimensions are $d_{in} = 4$ and $d_{out} = 1$, respectively. The components $\theta^*$ are set to either 0 or 3/4 according to the pattern depicted in Figure 7; the specifics of $f_{\psi^*}$ and $P_x^*$ are detailed in Appendix C. We result in a dataset of $35k$ input-output pairs $(x, y)$, 80% of which are used as training set, 10% as validation set, and the remaining 10% as test set. As predictive model family (2), we follow the same architecture of $f_{\psi^*}$ and $P_A^{\theta^*}$ ensuring that during all the experiments Assumption 3.1 is fulfilled. The model parameters are trained by optimizing the expected squared MMD in (9) with the rational quadratic kernel [Bińkowski et al., 2018].

## 6.1 GRAPH STRUCTURE LEARNING & OPTIMAL POINT PREDICTIONS

To test our method's ability to both calibrate the latent distribution and make optimal predictions, we train the model minimizing $\mathcal{L}^{dist}$ as described in Section 5.2.

Figure 2 reports the validation losses during training: MMD loss $\mathcal{L}^{dist}$ as in (6), MAE between the learned parameters $\theta$ and the ground truth $\theta^*$ as $\mathcal{L}^{cal}$ (5), and point-prediction loss $\mathcal{L}^{point}$ as in (4) with $\ell$ being the MAE. The results are averaged over 20 different model initializations and error bars report $\pm 1$ standard deviations from the mean. Results are reported with and without applying the variance reduction (Section 5.3), by training only parameters $\theta$ while freezing $\psi$ to $\psi^*$ (same setting of Theorem 5.2), and by joint training of both $\psi$ and $\theta$.

**Solving the joint learning problem**  Figure 2a shows that the training succeeded and the MMD loss $\mathcal{L}^{dist}$ approached its minimum (dotted line). Having minimized $\mathcal{L}^{dist}$, from Figure 2b we see that also the calibration of latent distribution $P_A^{\theta}$ was successful; in particular, the figure shows that the MAE on $\theta$ parameters ($N^{-2}\|\theta^* - \theta\|_1$) approaches zero as training proceeds (MAE < 0.04). Regarding the point predictions, Figure 2c confirms that $\mathcal{L}^{point}$ reached its minimum value; recall that optimal prediction MAE is not 0, as the target variable $y$ is random, and note that a learning rate reduction is applied at epoch number 5. The optimality of the point-prediction is supported also by the performance on separate test data and with respect to the MSE as point-prediction loss $\ell$. Moreover, we observe that calibration is achieved regardless of the variance reduction and whether or not parameters $\psi$ are trained. Lastly, Figure 4 shows the learned parameters $\theta$ of the latent distribution and the corresponding absolute discrepancy resulted from a (randomly chosen) training run.

**Optimization landscape of $\mathcal{L}^{point}$ and $\mathcal{L}^{dist}$**  In this experiment, we analyze the values of $\mathcal{L}^{point}(\psi^*, \theta)$ and $\mathcal{L}^{dist}(\psi^*, \theta)$ for different values of $\theta$. $\mathcal{L}^{point}$ is computed employing MAE as loss function $\ell$. Specifically, we let scalar $p$ vary from 1/2 to 1 and set all $\theta_{ij} = p$ for $i, j$ where $\theta_{ij}^* = 3/4$. Figure 1 reports the obtained results, highlighting an almost flat $\mathcal{L}^{point}$ for values

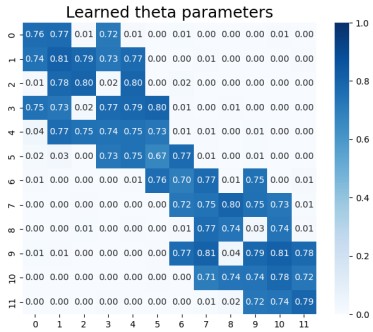

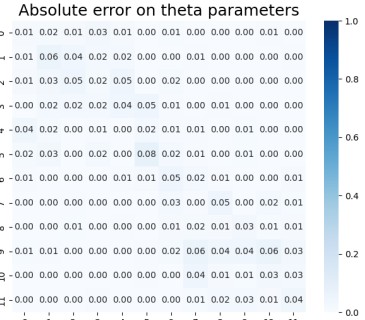

Figure 4: The learned parameters for the latent distribution corresponding to the stochastic adjacency matrix.

Figure 5: Absolute error made on the parameters of the latent distribution.

$\geq 0.725$. In contrast, $\mathcal{L}^{dist}$ displays a pronounced concave shape with a clear minimum around $\theta^*$ which suggests that calibration is easier when we minimize $\mathcal{L}^{dist}$ instead of $\mathcal{L}^{point}$.

Overall, we conclude that our approach is effective in solving the joint learning problem of calibrating the latent variable while producing optimal point predictions.

**Variance reduction effectiveness** Figures 2a, 2b and 2c demonstrate that the proposed variance reduction method (Section 5.2) yields notable advantages training speed up (roughly 50% faster). For this reason, the next experiments rely on variance reduction.

**Larger graphs** The theoretical results developed hold for any number of nodes $N$. However, the number of possible edges scales quadratically in the number of nodes. In Figure 3, we show all $\sim 15K$ parameters of the considered $P_A^\theta$ can be effectively learned even for relatively large graphs; the final MAE on $\theta$ parameters is 0.003. Note that for extremely large graphs the ratio between the number of free parameters in $\theta$ and the size of the training set can become prohibitive. In these cases, amortized learning of the edge probabilities is a potentially viable solution.

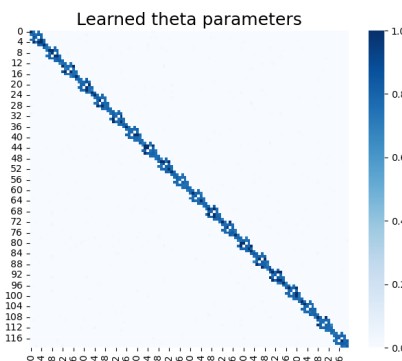

Figure 3: Learned $\theta$ parameters for a graph with $\sim 15K$ possible edges.

### 6.2 BEYOND ASSUMPTION 3.1

In this section, we empirically study whether Assumption 3.1 is restrictive in practical applications. Specifically, we consider different degrees of model mismatch between the system model in (1) and the approximating model in (2). Unless otherwise specified, we use the same dataset and experimental setup as described in Appendix C.1. Additional details and results are deferred to Appendix C.3.

**Perturbed $f_{\psi^*}$** As a first experiment, we train $P_A^\theta$ while keeping the parameters of the predictive function $f_\psi$ fixed to a random perturbation of the data-generating model $f^* = f_{\psi^*}$. A perturbed version of $f_\psi^*$ is built by uniformly drawing independent perturbation scalar values $\delta_i \sim \mathcal{U}[-\Psi, \Psi]$, one for each of parameter $\psi_i^*$ of $f_{\psi^*}$. Then, each parameter of GNN $f_\psi$ is given as $\psi_i = (1 + \delta_i)\psi_i^*$. Table 1 shows that the learned latent distribution remains reasonably calibrated, even when parameters can be modified up to 80%. In particular, the absolute error (AE) on parameters $\theta$ is under 10% on average and increases with $\Psi$. Finally, Figures 8-11

Table 1: Calibration of $P_A^\theta$ under varying levels of misconfiguration for predictive function $f_\psi$. Results are the mean $\pm$ 1 standard deviation assessed over 8 independent runs.

| Max pert. $\Psi$ | MAE on $\theta$ | Max AE on $\theta$ |
|---|---|---|
| 0 | $0.018 \pm 0.005$ | $0.12 \pm 0.01$ |
| 0.1 | $0.02 \pm 0.01$ | $0.12 \pm 0.01$ |
| 0.2 | $0.02 \pm 0.02$ | $0.14 \pm 0.03$ |
| 0.5 | $0.03 \pm 0.02$ | $0.20 \pm 0.12$ |
| 0.8 | $0.07 \pm 0.02$ | $0.36 \pm 0.08$ |

show the learned parameter vectors $\theta$ for randomly extracted runs and highlight that the maximum AE of Table 1 is observed only sporadically.

**Generic GNN as $f_\psi$** In this second experiment, we set $f_\psi$ to be a generic multilayer GNN which we jointly train with graph distribution $P_A^\theta$. We comment that model family $\{f_\psi\}$ does not include $f^*$, as $f^*$ uses L-hop adjacency matrices generated from the sampled adjacency matrix $A$, while the learnable $f_\psi$ relies on multiple nonlinear 1-hop layers; details on the model architecture are reported in Appendix C.3. Upon convergence, models achieved a MAE on $\theta < 0.11$ and $\mathcal{L}^{\text{point}} < 0.34$ using the MAE as loss function $\ell$ in (4); The performance is in line with results in Figure 2c and Table 1. At last, we note that because the GNN used adds self-loops, the diagonal elements of the adjacency matrix are learned as zero, resulting in a larger MAE on $\theta$ (see Figure 12). However, this does not impair the learning the off-diagonal $\theta_{ij}$ parameters (i.e., for $i \neq j$). Notably, in the worst-performing model, these off-diagonal parameters have a MAE of 0.05.

**Misconfigured $P_A^\theta$** Finally, we violate Assumption 3.1 by fixing $f_\psi = f^*$ and constraining some components of $\theta$ to incorrect values. Specifically, we force parameters $\theta_{i,j}$ for all edges $i, j$ associated with nodes with id 2 and 3 in Figure 6 to 0.25, instead of the correct value of $\theta_{i,j}^* = 0.75$ as in $P_A^*$. Results indicate that the free parameters in $\theta$ are learned appropriately. Notably, increased uncertainty is observed for spurious edges linking to nodes in the first node community (see Figure 6). This is expected given that nearly 60% of the edges in the community were significantly downsampled. Figures 13 and 14 in Appendix C.3 show the learned parameters from randomly selected runs.

## 7 CONCLUSIONS

Graph structure learning has emerged as a research field focused on learning graph topologies in support of solving downstream predictive tasks. Assuming stochastic latent graph structures, we are led to a joint optimization objective: (i) learning the correct distribution of the latent topology while (ii) achieving optimal predictions on the downstream task. In this paper, at first, we prove both positive and negative theoretical results to demonstrate that appropriate loss functions must be chosen to solve this joint learning problem. Second, we propose a sampling-based learning method that does not require the computation of the predictive likelihood. Our empirical results demonstrate that this approach achieves optimal point predictions on the considered downstream task while also yielding calibrated latent graph distributions.

Finally, we acknowledge that the proposed method requires sampling and processing multiple adjacency matrices for each input and, although the model and prediction accuracy is enhanced, a computation overhead is requested. We plan future research to explore the applicability of this method to other classes of neural networks beyond GNNs.

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

# APPENDIX

## A  PROOFS OF THE THEORETICAL RESULTS

### A.1  MINIMIZING $\mathcal{L}^{point}$ DOES NOT GUARANTEE CALIBRATION

In this section, we prove Proposition 4.1.

*Proof of Proposition 4.1.* Recall the definition of $\mathcal{L}^{point}$ in (4) using (3)

$$\mathcal{L}^{point}(\psi, \theta) = \mathbb{E}_x \Big[ \mathbb{E}_{y^* \sim P^*_{y|x}} \Big[ \ell \big(y^*, T \big[P^{\theta, \psi}_{y|x}\big]\big) \Big] \Big]$$

Given loss function $\ell$, $T$ is, by definition [Berger, 1990; Gneiting, 2011], the functional that minimizes

$$\mathbb{E}_{y^* \sim P^*_{y|x}} \Big[ \ell \big(y^*, T \big[P^*_{y|x}\big]\big) \Big]$$

Therefore, if $P^{\theta, \psi}_{y|x} = P^*_{y|x} \implies \mathcal{L}^{point}$ is minimal. If another distribution over $y$, namely, $P^{\psi', \theta'}_{y|x}$ parametrized by $\theta'$ and $\psi'$ satisfies:

$$T \Big[ P^{\psi', \theta'}_{y|x} \Big] = T \Big[ P^*_{y|x} \Big]$$

almost surely on $x$, then,

$$\mathcal{L}^{point}(\theta', \psi') = \mathbb{E}_x \Big[ \mathbb{E}_{y^* \sim P^*_{y|x}} \Big[ \ell \big(y^*, T \big[P^{\psi', \theta'}_{y|x}\big]\big) \Big] \Big]$$
$$= \mathbb{E}_x \Big[ \mathbb{E}_{y^* \sim P^*_{y|x}} \Big[ \ell \big(y^*, T \big[P^*_{y|x}\big]\big) \Big] \Big]$$

Thus, $P^{\psi', \theta'}_{y|x}$ minimizes $\mathcal{L}^{point}$.

Appendix A.2 discusses graph distributions where $T \big[P^{\psi', \theta'}_{y|x}\big] = T \big[P^*_{y|x}\big]$ but $P^{\psi', \theta'}_{y|x} \neq P^*_{y|x}$. We conclude that reaching the minimum of $\mathcal{L}^{point}(\psi, \theta)$ does not imply $P^{\psi, \theta}_{y|x} = P^*_{y|x}$.  $\square$

### A.2  MINIMIZING $\mathcal{L}^{point}$ DOES NOT GUARANTEE CALIBRATION: AN EXAMPLE WITH MAE

This section shows that $\mathcal{L}^{point}$ equipped with MAE as $\ell$ admits multiple global minima for different parameters $\theta$, even for simple models and $f_\psi = f^*$.

Consider a single Bernoulli of parameter $\theta^* > 1/2$ as latent variable $A$ and a scalar function $f^*(x, A)$ such that $f^*(x, 1) > f^*(x, 0)$ for all $x$. Given input $x$ the value of functional $T(P^*_{y|x})$ that minimizes

$$\mathbb{E}_{y \sim P^*_{y|x}} \Big[ \big|y - T \big[P^*_{y|x}\big]\big| \Big] = \theta^* \Big|f^*(x, 1) - T \big[P^*_{y|x}\big]\Big| + (1 - \theta^*) \Big|f^*(x, 0) - T \big[P^*_{y|x}\big]\Big|$$

is $T(P^*_{y|x}) = f^*(x, 1)$; this derives from the fact that range of $f^*$ is $\{f^*(x, 0), f^*(x, 1)\}$ and the likelihood of $f^*(x, 1)$ is larger than that of $f^*(x, 0)$.

Note that $T \big[P^*_{y|x}\big] = f^*(x, 1)$ for all $x$, therefore also $\mathcal{L}^{point}$ is minimized by such $T$. Moreover, $T \big[P^*_{y|x}\big]$ is function of $\theta^*$ and equal to $f^*(x, 1)$ for all $\theta > 1/2$. We conclude that for any $\theta \neq \theta^*$ distributions $P^{\theta, \psi}_{y|x}$ and $P^*_{y|x}$ are different, yet both of them minimize $\mathcal{L}^{point}$ if $\theta > 1/2$.

A similar reasoning applies for $\theta^* < 1/2$.

### A.3  MINIMIZING $\mathcal{L}^{dist}$ GUARANTEES CALIBRATION AND OPTIMAL POINT PREDICTIONS

This section proves Theorem 5.2 and a corollary of it.

*Proof of Theorem 5.2.* Recall from Equation (6) that

$$\mathcal{L}^{dist}(\theta) = \mathbb{E}_x\left[\Delta(P^*_{y|x}, P^\theta_{y|x})\right]$$

We start by proving that if $\mathcal{L}^{dist}(\theta, \psi) = 0 \implies \mathcal{L}^{point}(\theta, \psi)$ is minimal.

Note that $\mathcal{L}^{dist}(\theta, \psi) = 0$ implies that $\Delta(P^*_{y|x}, P^\theta_{y|x}) = 0$ almost surely in $x$. Then, by Assumption 5.1, $P^*_{y|x} = P^{\psi,\theta}_{y|x}$ almost surely on $x$ and, in particular, $T[P^*_{y|x}] = T[P^{\psi,\theta}_{y|x}]$, which leads to $\mathcal{L}^{point}(\psi, \theta)$ being minimal (Proposition 4.1).

We now prove that if $\mathcal{L}^{dist}(\theta, \psi^*) = 0 \implies \mathcal{L}^{cal}(\theta) = 0$.

From the previous step, we have that $\mathcal{L}^{dist}(\theta, \psi) = 0$ implies $P^*_{y|x} = P^{\psi,\theta}_{y|x}$ almost surely for $x \in I$. Under the assumption that $f_\psi = f_*$ and the injectivity of $f_*$ in such $x \in I$, for any output $y$ a single $A$ exists such that $f_*(x, A) = y$. Therefore, the probability mass function of $y$ equals that of $A$. Accordingly, $P^*_{y|x} = P^{\psi,\theta}_{y|x}$ implies $P^*_A = P^\theta_A$.

$\square$

We also prove a corollary of Theorem 5.2.

**Corollary A.1.** *Under Assumptions 3.1 and 5.1, if*

    *1. $\exists \bar{x} \in Supp(P^*_x) \subseteq \mathcal{X}$ such that $f^*(\bar{x}; \cdot)$ is injective,*

    *2. $f^*(x, A)$ is continuous in $\bar{x} \; \forall A \in \mathcal{A}$,*

*then*

$$\mathcal{L}^{dist}(\theta, \psi^*) = 0 \implies \begin{cases} \mathcal{L}^{point}(\theta, \psi^*) \text{ is minimal} \\ \mathcal{L}^{cal}(\theta) = 0. \end{cases}$$

The corollary shows that it is sufficient that $f^*$ is continuous in $x$ and there exists one point $\bar{x}$ where $f^*(\bar{x}, \cdot)$ is injective to meet theorem's hypothesis $\mathbb{P}_{x \sim P^*_x}(I) > 0$; we observe that, as $\mathcal{A}$ is discrete, the injectivity assumption is not as restrictive as if the domain were continuous.

*Proof.* As $\mathcal{A}$ is a finite set, the minimum $\bar{\epsilon} = \min_{A, A' \in \mathcal{A}} \|f^*(\bar{x}, A) - f^*(\bar{x}, A')\| > 0$ exists and, by the injectivity assumption, is strictly positive.

By continuity of $f^*(\cdot, A)$, for every $\epsilon < \frac{1}{2}\bar{\epsilon}$ there exists $\delta$, such that for all $x \in B(\bar{x}, \delta)$ we have $\|f^*(\bar{x}, A) - f^*(x, A)\| < \epsilon$. It follows that, $\forall x \in B$,

$$\begin{aligned}
\|f^*(x, A) &- f^*(x, A')\| \\
&\geq \|f^*(\bar{x}, A) - f^*(\bar{x}, A')\| - \|f^*(\bar{x}, A) - f^*(x, A)\| - \|f^*(\bar{x}, A') - f^*(x, A')\| \\
&\geq \|f^*(\bar{x}, A) - f^*(\bar{x}, A')\| - 2\epsilon \\
&\geq \|f^*(\bar{x}, A) - f^*(\bar{x}, A')\| - \bar{\epsilon} > 0.
\end{aligned}$$

Finally, as $\bar{x} \in \text{Supp}(P^*_x)$ and $B(\bar{x}, \delta) \subseteq I$, we conclude that

$$\mathbb{P}_x(I) \geq \mathbb{P}_x(B(\bar{x}, \delta)) > 0,$$

therefore, we are in the hypothesis of Theorem 5.2 and can conclude that

$$\mathcal{L}^{dist}(\theta, \psi^*) = 0 \implies \begin{cases} \mathcal{L}^{point}(\theta, \psi^*) \text{ is minimal} \\ \mathcal{L}^{cal}(\theta) = 0. \end{cases}$$

$\square$

## A.4 INJECTIVITY HYPOTHESIS FOR GRAPH NEURAL NETWORKS

Now, we show that hypothesis $\mathbb{P}_{x \sim P_x^*}(I) > 0$ of Theorem 5.2 is always met for certain families of graph neural networks.

**Proposition A.2.** *Consider a 1-layer GNN of the form $f^*(x, A) : \sigma(Ax) = y$, with $x, y \in \mathbb{R}^N$ and nonlinear bijective activation function $\sigma$. If the support $Supp(P_x^*)$ of $x$ contains any ball $B$ in $\mathbb{R}^N$ then $\mathbb{P}_{x \sim P_x^*}(I) > 0$.*

To prove Proposition A.2, we rely on following lemma.

**Lemma A.3.** *Given $g(x, a) = ax$ with $a \in \{0, 1\}^{1 \times N}$ and $x \in \mathbb{R}^{N \times 1}$. Let $I_g = \{x : g(x, a) \text{ is injective in } a\} \subseteq \mathcal{X}$ be the set of points $x \in \mathcal{X}$ such that map $a \mapsto g(x, a)$ is injective. The following implication holds:*

$$x \notin I_g \iff \exists \delta \neq \mathbf{0} \in \{-1, 0, 1\}^{1 \times N} \text{ s.t. } \delta \perp x. \tag{14}$$

*Proof.* We prove the two implications separately.

($\implies$) If $x \notin I_g$, then there exist $a', a'' \in \{0, 1\}^{1 \times N}$ with $a' \neq a''$ such that $a'x = a''x$. This implies that $(a' - a'')x = 0$. Defining $\delta$ as $(a' - a'')$, we have proven that there exist $\delta \neq \mathbf{0} \in \{-1, 0, 1\}^{1 \times N}$ such that $\delta x = 0$, i.e., $\delta \perp x$.

($\impliedby$) Assume that $\exists \, \delta \neq \mathbf{0} \in \{-1, 0, 1\}^{1 \times N}$ such that $\delta \perp x$. Each component $\delta_i$ of $\delta$ can be written as the difference between two values $a_i', a_i'' \in \{0, 1\}$. As $\delta \neq \mathbf{0}$ then there exists at least one index $j \in \{1, \ldots, N\}$ such that $a_j' \neq a_j''$. This implies that $\exists \, a', a'' \in \{0, 1\}^{1 \times N}$ with $a' \neq a''$ s.t. $(a' - a'')x = 0$, which implies that $x \notin I_g$.

$\square$

*Proof of Proposition A.2.* We begin by considering the projection $\bar{g}(x, a) = ax$ with $a \in \{0, 1\}^{1 \times N}$ and $x \in R^N$. Then we extend to $A \in \{0, 1\}^{N \times N}$ and to nonlinear functions.

Let $I_{\bar{g}}^C = \mathbb{R}^N \setminus I_{\bar{g}}$ be the complement in $\mathbb{R}^N$ of $I_{\bar{g}}$. Recalling Lemma A.3 and its notation, we have $3^N - 1$ possible $\delta$, defining a collection of $(3^N - 1)/2$ hyperplanes of vectors $x$ perpendicular to at least one $\delta$; set $I_{\bar{g}}^C$ is the union of such a finite collection of hyperplanes. By hypothesis, $Supp(P_x^*)$ contains a ball $B \in \mathbb{R}^N$, therfore $Supp(P_x^*) \not\subset I_{\bar{g}}^C$ and $\mathbb{P}_{x \sim P_x^*}(I_{\bar{g}}^C) < 1$. We conclude that $\mathbb{P}_{x \sim P_x^*}(I_{\bar{g}}) = 1 - \mathbb{P}_{x \sim P_x^*}(I_{\bar{g}}^C) > 0$.

A similar result is proven for $\bar{G}(x, A) = Ax$ with $A \in \{0, 1\}^{N \times N}$. In fact, $\bar{G}$ is a stack of $N$ functions $\bar{g}$ above and $I_{\bar{G}} = I_{\bar{g}}$. Finally, composing injective function $G$ with injective function $\sigma$ leads to function $g(x, A) = \sigma(G(x, A))$ being injective in $A$ for the same points $x$ for which $G$ is injective, thus proving the proposition. $\square$

## B ESTIMATION OF OPTIMAL $\beta_1$ AND $\beta_2$

Here we show that, when reducing the variance of the SFE via control variates in (12), the best $\beta_1$ and $\beta_2$ can be approximated by

$$\tilde{\beta}_1 = \mathop{\mathbb{E}}_{\substack{x \sim P_x^* \\ A_1, A_2 \sim P_A^\theta}} \Big[ \kappa \left( f_\psi(x, A_1), f_\psi(x, A_2) \right) \Big], \qquad \tilde{\beta}_2 = \mathop{\mathbb{E}}_{\substack{(x, y^*) \sim P_{x,y}^* \\ A \sim P_A^\theta}} \Big[ \kappa \left( y^*, f_\psi(x, A) \right) \Big], \tag{15}$$

Consider generic function $L(A)$ depending on a sample $A$ of a parametric distribution $P_A^\theta(A)$ and the surrogate loss $\tilde{L}(A)$ in (11), i.e.,

$$\tilde{L}(A) = L(A) - \beta \Big( h(A) - \mathbb{E}_{A \sim P^\theta}[h(A)] \Big); \tag{16}$$

This choice is not new in the literature [Sutton et al., 1999; Mnih et al., 2016] where $\beta$ is often referred to as *baseline*. The 1-sample MC approximation of the loss becomes

$$\nabla_\theta \mathbb{E}_{A \sim P^\theta}[L(A)] \approx \tilde{L}(A') \nabla_\theta \log P^\theta(A') = (L(A') - \beta) \nabla_\theta \log P^\theta(A'), \tag{17}$$

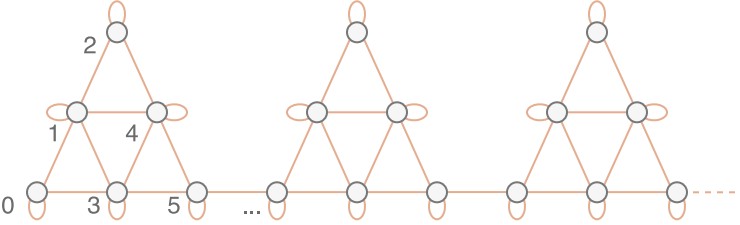

Figure 6: The adjacency matrices used in this paper are sampled from this graph. Each edge in orange is independently sampled with probability $\theta^*$. In the picture, 3 communities of an arbitrarily large graph are shown.

with $A'$ sampled from $P_A^\theta$. The variance of the estimator is

$$\mathbb{V}_{A \sim P^\theta}\left[(L(A) - \beta)\nabla_\theta \log P^\theta(A)\right] = \mathbb{V}_{A \sim P^\theta}\left[L(A)\nabla_\theta \log P^\theta(A)\right] +$$

$$+ \beta^2 \mathbb{E}_{A \sim P^\theta}\left[\left(\nabla_\theta \log P^\theta(A)\right)^2\right] - 2\beta \mathbb{E}_{A \sim P^\theta}\left[L(A)\left(\nabla_\theta \log P^\theta(A)\right)^2\right] \quad (18)$$

and the optimal value $\beta$ that minimizes it is

$$\tilde{\beta} = \frac{\mathbb{E}_{A \sim P^\theta}\left[L(A)\left(\nabla_\theta \log P^\theta(A)\right)^2\right]}{\mathbb{E}_{A \sim P^\theta}\left[\left(\nabla_\theta \log P^\theta(A)\right)^2\right]}. \quad (19)$$

If we approximate the numerator with $\mathbb{E}[L(A)]\mathbb{E}[(\nabla_\theta \log P^\theta(A))^2]$, we obtain that $\tilde{\beta} \approx \mathbb{E}[L(A)]$. By substituting $L(A)$ with the two terms of (10) we get the values of $\beta_1$ and $\beta_2$ in (15).

We experimentally validate the effectiveness of this choice of $\beta$ in Section 6.

## C  FURTHER EXPERIMENTAL DETAILS

### C.1  DATASET DESCRIPTION AND MODELS

In this section, we describe the considered synthetic dataset, generated from the system model (1). The latent graph distribution $P_A^*$ is a multivariate Bernoulli distribution of parameters $\theta_{ij}^*$: $P_A^* \equiv P_{\theta^*}(A) = \prod_{ij} \theta_{ij}^{*A_{ij}}(1 - \theta_{ij}^*)^{(1-A_{ij})}$. The components of $\theta^*$ are all null, except for the edges of the graph depicted in Figure 6 which are set to $3/4$. A heatmap of the adjacency matrix can be found in Figure 7.

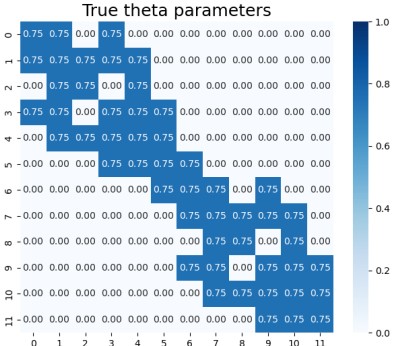

Figure 7: $\theta_{ij}^*$ parameters for each edge of the latent adjacency matrix. Each square corresponds to an edge, and the number inside is the probability of sampling that edge for each prediction.

Table 2: Table of the parameters used to generate the synthetic dataset.

| | |
|---|---|
| $\theta^*$ | 0.75 |
| $\sigma_x$ | 1.5 |
| $N$ | 12 |
| $d_{in}$ | 4 |
| $d_{out}$ | 1 |
| $\psi_1^*$ | $[-0.2, 0.4, -0.8, 0.6]$ |
| $\psi_2^*$ | $[-0.3, 0.8, 0.2, -0.7]$ |

Regarding the GNN function $f^*$, we use the following system model:

$$\begin{cases} y = f_{\psi^*}(A, x) = \tanh\left(\sum_{l=1}^{L} \mathbb{1}[A^l \neq 0]x\psi_l^*\right) \\ A \sim P_{\theta^*}(A) \end{cases} \tag{20}$$

where $\mathbb{1}[\cdot]$ is the element-wise indicator function: $\mathbb{1}[a] = 1 \iff a$ is true. $x \in \mathbb{R}^{N \times d_{in}}$ are randomly generated inputs: $x \sim \mathcal{N}(0, \sigma_x^2 \mathbb{I})$. $\psi_l^* \in \mathbb{R}^{d_{out} \times d_{in}}$ are part of the system model parameters. We summarize the parameters considered in our experiment in Table 2.

The approximating model family (2) used in the experiment is the same as the data-generating process, with all components of parameter vectors $\theta$ and $\psi$ being trainable. The squared MMD discrepancy is defined over Rational Quadratic kernel [Bińkowski et al., 2018]

$$\kappa(y', y'') = \left(1 + \frac{\|y' - y''\|_2^2}{2\alpha\sigma^2}\right)^{-\alpha} \tag{21}$$

of parameters $\sigma = 0.7$ and $\alpha = 0.02$.

The model is trained using Adam optimizer [Kingma & Ba, 2014] with parameters $\beta_1 = 0.6$, $\beta_2 = 0.95$. Where not specified, the learning rate is set to $0.1$ and decreased to $0.01$ after 5 epochs. We grouped data points into batches of size 128. Initial values of $\theta$ are independently sampled from the $\mathcal{U}(0.25, 0.35)$ uniform distribution.

## C.2 DESCRIPTION OF THE EXPERIMENT IN SECTION 4

In this experiment, we generate 512 data points using the system model described in Appendix C.1. We construct a model identical to the system model, except that $\theta_{ij} = p$ for all $i, j$ where $\theta_{i,j}^* = 0.75$ and 0 elsewhere. We vary scalar $p$ from $0.5$ to $1$ with steps of $0.025$. Therefore, only the model with $p = 0.75$ is identical to the data-generating model.

For each input $x$ in the dataset, a point prediction is produced by sampling $N_{adj} = 32$ adjacency matrices and computing the median. This approach allows to estimate $\mathcal{L}^{point}$ using the MAE as loss function $\ell$, as depicted by the red points in Figure 1, for different values of $\theta$. For comparison purposes, we estimate $\mathcal{L}^{dist}$ using the maximum mean discrepancy as proposed in Section 5.

## C.3 ADDITIONAL DETAILS OF SECTION 6.2

We present here additional Figures discussed in Section 6.2.

**Fixed perturbed** $f_\psi$    Figures in this paragraph correspond to the experiment where the processing function $f_\psi$ is fixed on a perturbed version of $f^*$. Figures $8 - 11$ correspond to runs with increasing perturbation factor $\Psi$.

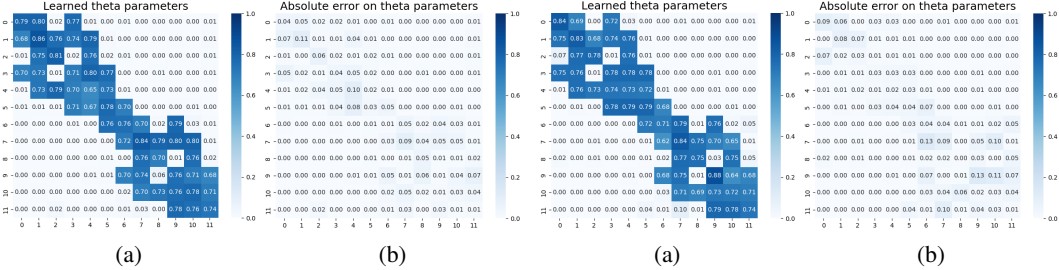

(a)      (b)      (a)      (b)

Figure 8: Learned $\theta_{ij}$ parameters (a) and Absolute Error (b) for maximum perturbation factor $\Psi$ of 10%.

Figure 9: Learned $\theta_{ij}$ parameters (a) and Absolute Error (b) for maximum perturbation factor $\Psi$ of 20%.

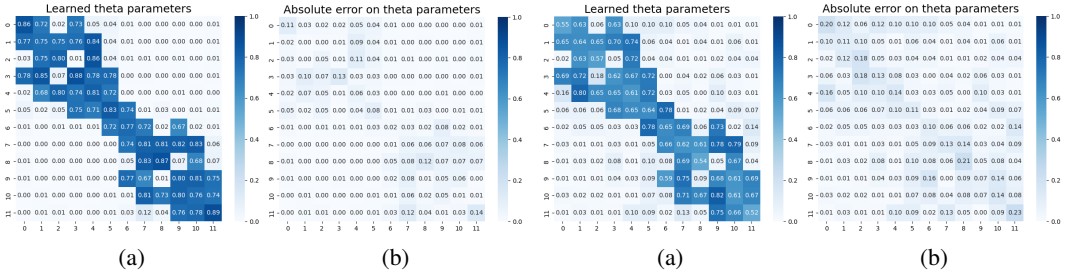

(a)      (b)      (a)      (b)

Figure 10: Learned $\theta_{ij}$ parameters (a) and Absolute Error (b) for maximum perturbation factor $\Psi$ of 50%.

Figure 11: Learned $\theta_{ij}$ parameters (a) and Absolute Error (b) for maximum perturbation factor $\Psi$ of 80%.

**Generic GNN as $f_\psi$** To evaluate our approach in a more realistic setting, we use a generic GNN as $f_\psi$. Specifically, we implement GNNs from [Morris et al., 2019] with varying numbers of layers and layer sizes. It is important to note that the GNN implementation includes self-loops, which prevents the diagonal elements from being correctly learned. However, this does not impede our method from learning the remaining edges accurately.

Table 3 presents the network configurations and whether they successfully converged to the ground truth distribution. Since diagonal elements artificially inflate the MAE for $\theta$, we consider a model to have converged if the final MAE on $\theta$ is less than 0.11.

Table 3: Network configurations and corresponding convergence results.

| Layers dimensions | Convergence |
| --- | --- |
| $[4, 1]$ | **x** |
| $[4, 1, 1]$ | **x** |
| $[4, 2, 1]$ | ✓ |
| $[4, 8, 1]$ | ✓ |
| $[4, 8, 2, 1]$ | ✓ |
| $[4, 16, 8, 1]$ | ✓ |
| $[4, 32, 8, 1]$ | ✓ |
| $[4, 64, 8, 1]$ | ✓ |
| $[4, 64, 16, 1]$ | ✓ |
| $[4, 64, 32, 1]$ | ✓ |
| $[4, 8, 8, 4, 1]$ | ✓ |

Most of the models successfully converged, except those with high bias. This demonstrates that our method is effective even beyond Assumption 3.1. In Figure 12 we show the learned parameters of $P_A^\theta$ for a randomly extracted run.

**Misconfigured $P_A^\theta$** Figures 13 and 14 correspond to the experiment where some $\theta_{ij}$ values of $P_A^\theta$ are fixed at incorrect values, while the processing function $f_\psi$ is fixed to the true one. In the community affected by the perturbation, free $\theta_{ij}$ values tend to be sampled more frequently to compensate for the downsampling imposed by the perturbation. Interestingly, all the edges with at least one edge in the second community (75% of the edges) appear unaffected by the perturbation.

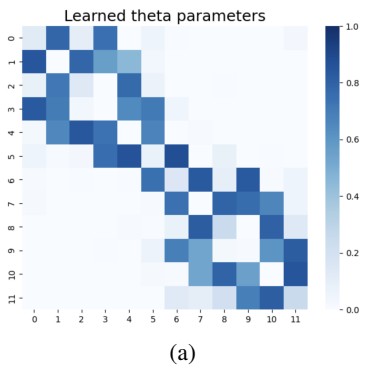
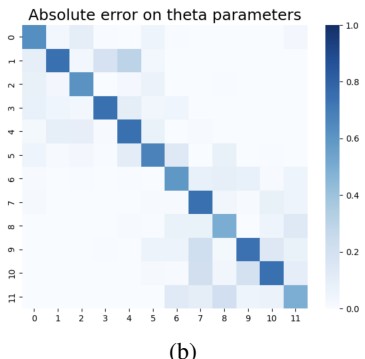

(a)                                      (b)

Figure 12: (a) Learned $\theta_{ij}$ parameters when the parametric processing function $f_\psi$ is a generic GNN as presented in [Morris et al., 2019] and (b) Absolute Error made with respect to true parameters $\theta^*_{ij}$. As self-loops are deterministically added by the network, the diagonal elements should not be considered.

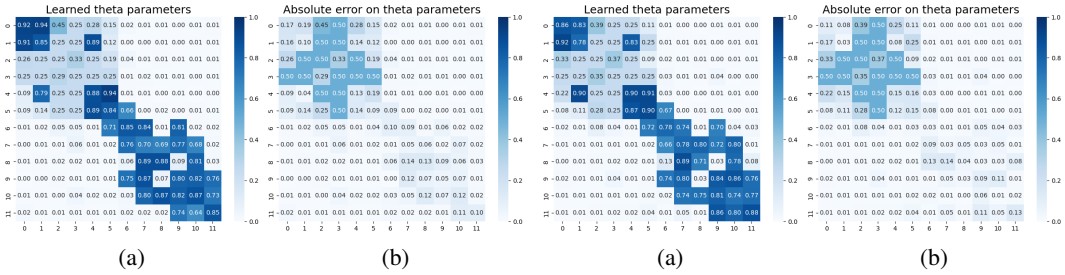

(a)                  (b)                            (a)                  (b)

Figure 13: Learned $\theta_{ij}$ parameters (a) and Absolute Error (b) for misconfigured $P^\theta_A$

Figure 14: Learned $\theta_{ij}$ parameters (a) and Absolute Error (b) for misconfigured $P^\theta_A$

## C.4 Compute resources and open-source software

The paper's experiments were run on a workstation with AMD EPYC 7513 processors and NVIDIA RTX A5000 GPUs; on average, a single model training terminates in a few minutes with a memory usage of about 2GB.

The developed code relies on PyTorch [Paszke et al., 2019] and the following additional open-source libraries: PyTorch Geometric [Fey & Lenssen, 2019], NumPy [Harris et al., 2020] and Matplotlib [Hunter, 2007].

