# OpenReview forum: "Learning Latent Graph Structures and their Uncertainty"
_ICLR.cc/2025/Conference — Submitted to ICLR 2025_

### Official Review · Reviewer_yvut · 2024-10-27

**Soundness:** 4
**Presentation:** 3
**Contribution:** 2
**Rating:** 6
**Confidence:** 3

**Summary:**

(After reading the revised version, I increased my score from 5 to 6.)

This paper addresses the problem of jointly learning the latent adjacency matrix $A$ and a node inference task via a function $f(A, x)$. It considers a family of predictive models $P_{y|x}^{\theta, \psi}$, where one component learns the distribution $P(A)$ and another models the inference task $f$. The paper shows that optimizing a pointwise loss does not necessarily ensure proper learning of the distribution over $P(A)$. To address this, the authors propose a new loss formulation based on a distribution dissimilarity metric that guarantees learning both the latent graph structure and the inference function for the considered class of functions and settings. The methodology provides one example of how to optimize such a loss by choosing Maximum Mean Discrepancy (MMD) as the dissimilarity metric. The authors then develop a method to optimize the proposed MMD loss and provide empirical experiments to show that their approach successfully learns both the distribution $P(A)$ and the inference function $f$

**Strengths:**

1. The paper is well-structured and clearly written, with a logical flow that guides the reader through the problem, methodology, and results on the synthetic experiment.

2. The paper provides a comprehensive treatment of the problem, carefully addressing its assumptions and hypotheses.  It thoroughly discusses the implications and limitations of each assumption and hypothesis, and it empirically verifies what happens when the assumptions are violated.

3. The proposed loss function is principled, built on a solid theoretical foundation.

**Weaknesses:**

1. Motivation.
The motivation for the joint learning task is not clearly articulated. The paper’s main result highlights cases where successfully solving one task does not imply solving an unrelated task (i.e., learning the distribution of a latent random variable). However, the primary cited motivation for learning the graph distribution is to improve performance on a downstream task. If the downstream task can already be solved, it is unclear why learning the distribution over a latent graph would then be necessary. In summary, the authors should provide a practical setting where their proposed joint learning task and loss function would be called for.

2. Contribution. The paper's main theoretical contribution appears to be the observation that optimizing for task $X$ (achieving optimal point prediction) does not inherently lead to solving subtask $Y$ (learning the distribution over the latent adjacency matrix). This outcome is somewhat expected, given that the problem formulation lacks any dependencies linking subtask $Y$ to task $X$. A more specific formulation that establishes an explicit connection between the two tasks might provide a stronger foundation for the contribution.

3. Scope of the experiment. The experiments are limited to a small synthetic dataset with a small number of nodes ($N = 12$), a fixed feature dimension ($d = 4$), and fixed graph sparsity. The paper does include an experiment with more nodes ($N = 116$); however, the authors note that beyond this number, the number of free parameters becomes prohibitive. A more extensive empirical section showing how training with the proposed loss scales with the number of nodes, the feature dimension, and the graph sparsity (of the ground truth) would strengthen the presentation and provide more insight into the limits and applications of the proposed loss.

Relevant citations
Xingyue Pu, Tianyue Cao, Xiaoyun Zhang, Xiaowen Dong, and Siheng Chen. Learning to learn
graph topologies. In Advances in Neural Information Processing Systems, pages 4249–4262,
2021.
Ruoyu Li, Sheng Wang, Feiyun Zhu, and Junzhou Huang. Adaptive graph convolutional neural
networks. In Proceedings of the AAAI Conference on Artificial Intelligence, 2018.

Antonio Ortega, Pascal Frossard, Jelena Kovacevic, José MF Moura, and Pierre Vandergheynst.
Graph signal processing: Overview, challenges, and applications. Proceedings of the IEEE,
106(5):808–828, 2018.

Minor

1. (68) typo
2. (128) Missing end of sentence
3. (275) typo substituhte

**Questions:**

1. Why is the problem of learning the distribution of a latent random variable (the adjacency matrix) relevant in cases where the inference task can be perfectly solved? Some settings where sampling the graph, evaluating edge probability or explainability should be put forward to better ground the contribution.

2. Where does $P_x*$ comes from? There is no discussion around $X$ being a random variable in the problem setting, and it's relation (if any) to $p(A)$. Currently, it may seems like $A$ and $X$ are assumed to be independent, which seems unlikely?

3. Why is there no baselines presented in the experiment section? For example, Anees Kazi et al. (2022) could have been presented to highlight the importance of the proposed joint learning formulation.

4. Aren't most GNN architecture not injective and would violate the injective constraint from Theorem 5.2?

---

> ### Author Response · Authors · 2024-11-20
>
> We appreciate the time and effort taken to provide thoughtful comments and additional references. Thank you for spotting the typos.
>
> **Motivation** and **Q1**.
> Please allow us to clarify that our primary motivation for learning the latent graph distribution (task B) is not to improve the performance on the downstream task (task A).
> However, while task B is not necessary for solving task A, there are multiple reasons for addressing the two tasks jointly. Learning appropriate models of the data-generating process (calibrating the latent variable without compromising prediction accuracy) can provide valuable insights into the modeled environment.
> Uncertainty quantification improves explainability and interpretability, ultimately enabling more informed decision-making. Examples of applications are found in the study of infection and information spreading, and biological systems, e.g.,
> see (Gomez Rodriguez et al 2013), (Lokhov 2016), (Deleu et al. 2022).
>
>
> **Contribution**.
> There may be some misunderstanding here. The mentioned piece of contribution (i.e., "optimizing for task $X$ (...) does not inherently lead to solving subtask $Y$ (...).") is just the initial part of our contributions.
> We also would like to point out that we do not perceive task $Y$ (learning the distribution over the latent adjacency matrix) as a subtask, as it is not necessary for solving task $X$.
> Our main contribution is showing that it is possible to solve both tasks jointly (Theorem 5.2). In particular, we provide sufficient conditions that grant it and we show that these conditions can be mild for graphs and GNNs.
> Building on this, we developed a theoretically grounded and empirically validated learning approach (Sections 5 and 6).
>
> Explicit connections between tasks $X$ and $Y$ are established and formulated as the four implications appearing in Theorem 5.2 and Proposition 4.1.
> We believe these are strong results that suggest that loss functions as defined in Equation 5 may, in general, be preferred.
>
>
>
> **Scope of experiments**.
>
> The choice of $N = 116$ in our experiments does not represent an upper limit of our method. What we meant to say with "Note that for extremely large graphs the ratio between the number of free parameters in $\theta$ and the size of the training set can become prohibitive." (line 448) is that the learning problem in general can get ill-posed as the ratio grows. Therefore it is a potential issue for any GSL method. The exploration of extreme scenarios of a huge number of nodes should be tackled from a general GSL perspective and, in our opinion, out of this paper's scope. So, while very interesting, we defer it to future research.
>
> Additionally, our method is computationally more scalable compared to approaches relying on dense (non-strictly binary) adjacency matrices, as both the forward and backward computations - in our approach - can remain sparse.
> Please, refer to lines 341-345 and Cini et al. (2023) where it has been empirically validated. Finally, we comment that increasing the feature dimension $d$ provides more training signal, thus it is expected to facilitate learning.
>
>
> **Q1**.
> See above.
>
> **Q2**.
> $P_x^*$ is the data distribution of the inputs $x$ as produced by the data-generating process (data collection). There is no particular restriction on $x$, which could be modeled as a continuous, discrete, or even mixed random variable. In this paper, we focused on independent $A$ and $x$ to make the empirical validation and visualization more practical. Nonetheless, modeling $A\sim P_{A|x}$ is indeed possible, e.g., by expressing the distribution parameters $\theta$ of current $P^\theta_A$ in Eq. 2 as input-dependent: $\theta=g_\vartheta(x)$ with $\vartheta$ a new set of learnable parameters.
>
>
> **Q3**.
> To our knowledge, no previous work was designed to solve this joint task. As we have proven, existing methods can succeed or fail depending on the loss they consider. For instance, the mentioned work by Kazi et al. (2022) adopts a type of loss function for which calibration is not guaranteed.
> In summary, this paper focuses on training losses and shows that output distribution losses (Equation 6) are more appropriate than point-prediction losses (Equation 4), regardless of the considered model.
>
> **Q4**.
> Please note that the injectivity we refer to in Theorem 5.2 concerns the function $A\mapsto f^*(x, A)$, rather than the usual $x\mapsto f^*(x, A)$. Indeed, while the latter is often not injective, the former can be. In Appendix A.4, we demonstrate that the injectivity assumption can hold robustly. Furthermore, as noted in lines 236–242, the function does not need to be injective for all input points $x$ if $f^*$ is continuous, but only for one point.

---

> > ### Author Response · Authors · 2024-11-20
> >
> > **References**
> >
> > Gomez Rodriguez, Manuel, Jure Leskovec, and Bernhard Schölkopf. "Structure and dynamics of information pathways in online media." Proceedings of the sixth ACM international conference on Web search and data mining. 2013.
> >
> > Lokhov, Andrey. "Reconstructing parameters of spreading models from partial observations." Advances in Neural Information Processing Systems 29 (2016).
> >
> > Deleu, Tristan, et al. "Bayesian structure learning with generative flow networks." Uncertainty in Artificial Intelligence. PMLR, 2022.
> >
> > Cini, Andrea, Daniele Zambon, and Cesare Alippi. "Sparse graph learning from spatiotemporal time series." Journal of Machine Learning Research 24.242 (2023): 1-36.
> >
> > Kazi, Anees, et al. "Differentiable graph module (dgm) for graph convolutional networks." IEEE Transactions on Pattern Analysis and Machine Intelligence 45.2 (2022): 1606-1617.

---

> > > ### Comment · Reviewer_yvut · 2024-11-29
> > >
> > > After reading the updated version, my main concern about motivation has been addressed. I will increase my score.

---

### Official Review · Reviewer_YZqH · 2024-10-30

**Soundness:** 3
**Presentation:** 3
**Contribution:** 3
**Rating:** 6
**Confidence:** 3

**Summary:**

This paper aims to calibrate the latent distribution of graphs for predicting outcomes of interacting entities. To this end, the paper considers a dissimilarity measure, which can be f-divergence, Stein variational gradients, or maximum mean discrepancy, between two predictive distributions. Then, the paper demonstrates that minimizing this measure ensures success of calibration under certain conditions where conventional point-wise approaches fail. For practical implementation, the dissimilarity measure is considered with maximum mean discrepancy computed over finite samples. The experiments focus on validating the proposed theorem.

**Strengths:**

- This paper first explores calibrating graph structure learning by minimizing the dissimilarity measure between two distributions.
- The theoretical analysis shows how the proposed method can be beneficial in cases, such as when  $f^{\ast} = f_{\psi}$, where the proposed method succeeds while conventional methods fail.
- The authors show that proposed method can be useful by incorporating variance reduction techniques and addressing the complexity.
- The experiments validate the theorems from several perspectives.

**Weaknesses:**

I have no concerns about the methods or their theorem. However, my overall concerns stem from the experiments.

- **Lack of baselines.** The experiments only provide the outputs of the proposed method and validate the theorem. However, they do not provide experiments for showing how conventional approaches fail in contrast, such as their shortcomings in estimating the underlying graph distribution. This limits the emphasis on the advantages of the proposed method.

- **Lack of benchmarks.** The experiments focus on synthetic benchmarks with limited scope, which limits highlighting the benefits of the proposed method. Are there no conventional benchmarks, such as node classification benchmarks, that could better highlight the benefits in practice?

- **Lack of uncertainty analysis.** In my understanding, the ultimate goal is to model output $y$ uncertainty by calibrating the latent distribution for graphs. However, the experiments lack analysis of the capability to model this uncertainty.

**Questions:**

See weakness.

---

> ### Author Response · Authors · 2024-11-20
>
> We appreciate your feedback. Please allow us to address your questions and comments.
>
> **Baselines**.
> Please note that the focus of this paper is mainly on the type of loss function to optimize, which appears to be fundamental for successfully solving both tasks. Therefore, existing methods can succeed or fail depending on the loss they consider. Besides, none of them were designed to solve this joint task, to our knowledge.
> What we show is that output distribution losses (Equation 6) are more appropriate than point-prediction losses (Equation 4), and this holds regardless of the considered model as long as Assumption 3.1 is met.
> Finally, although we show that using the MMD and a score function gradient estimator is an appropriate choice, we are not claiming its superiority over other methods optimizing an output distribution loss.
>
> **Benchmarks**.
> Unfortunately, to our knowledge, no dataset provides the ground-truth latent distribution of the graph structure (i.e., $P_A^*$). In the absence of such information, we have no way of evaluating the quality of the learned $P_A^\theta$.
> For this reason, we constructed a synthetic dataset to validate our claims and approach.
>
> **Uncertainty analysis**.
> It seems there may have been a misunderstanding here. Our primary goal is not to model the uncertainty of the output $y$ by calibrating the latent variable $A$. Instead, we aim at calibrating the latent variable $A$ itself.
> The problem we are addressing is inherently more complex both to solve and evaluate; this is due to the absence of ground truth information about the unknown variable $A$ we intend to learn in real-world scenarios.
> Our theoretical results address this challenge by providing learning guarantees.

---

> ### Comment · Reviewer_YZqH · 2024-11-22
>
> Thank you for the response. In my understanding, this paper proposes a method that primarily focuses on better capturing underlying graph structures while predicting results related to the graph. Although I cannot ensure the practical utility of this method in the real world tasks, the choice of this method may be necessary (compared to existing approaches) for explainability and interpretability, such as understanding underlying biological interactions for determining the output. Thus, I have decided to increase my score from 5 to 6.

---

### Official Review · Reviewer_fT1o · 2024-11-01

**Soundness:** 3
**Presentation:** 3
**Contribution:** 3
**Rating:** 8
**Confidence:** 3

**Summary:**

This paper addresses the Graph Structure Learning (GSL) problem, sometimes referred to as the ‘Latent Graph Learning’ problem, which concerns co-learning the graph structure and GNN weights for downstream prediction. The paper reveals issues with point-predictor losses with regards to the learning of the latent graph structure, and proposes a sampling based approach to overcome such issues.

**Strengths:**

I believe this work to be the first to consider the calibration aspect of the latent graph structure, which is an interesting point.

A primary supporting argument for GSL methods is that the learned graph is itself useful for interpretability purposes. But most (all?) approaches make little effort to (i) show an instance where this is actually useful beyond a nice visualization or (ii) discuss why we should believe this underlying graph corresponds to some real object.

This work seems to address address (ii).

**Weaknesses:**

**Motivation**

As best I can understand, the paper attempts to address (ii), but without addressing (i), i.e. the motivation of the entire effort. Why should anyone care in the first place?

The authors attempt to provide high-level applications, e.g., “Examples include e.g., social interactions where links can intermittently be present, traffic flows affected by road closures and temporary detours, and adaptive communication routing. It follows that a probabilistic framework is appropriate to accurately capture the uncertainty in the learned relations whenever randomness affects the graph topology.”

But it is not clear to me how GSL is/would be actually used in these applications, why uncertainty over such an inferred latent graph is important in these applications, nor why calibrated uncertainty measures of the latent graph are important.

&nbsp;


**Gaps in Related Work**

There are large gaps in related work on graph structure learning, e.g. using unrollings and/or Bayesian Neural Network. For recent work see Graph Structure Learning with Interpretable Bayesian Neural Networks, which addresses learning graphs with uncertainty estimates. Link at the bottom.

&nbsp;


**Extension to Real Data**

As the underlying network topology is rarely ever observed, how do we evaluate whether our estimate of it (along with the corresponding uncertainty/calibration) is good in a real data setting? The only way I can think of is by using it to make downstream predictions on labels we do observe, i.e. marginalizing it out and evaluating the resulting posterior predictive.


&nbsp;


**Minor Comments**

In Eqn (1) and (2) it may make sense to place A \sim P^{\theta)_A above \hat{y} = f_{\psi}(x, A). This is more standard in the probabilistic community which often views this model as a data-generating process:  A must be sampled first before it can be used as an input for f.

&nbsp;

Graph Structure Learning with Interpretable Bayesian Neural Networks, https://openreview.net/forum?id=2noXK5KBbx

**Questions:**

Listed in my above comments, namely addressing motivation (if the downstream prediction is the same, why do we care at all about GSL producing high fidelity graph estimates?) and extensions to real data.

---

> ### Author Response · Authors · 2024-11-20
>
> Thank you for your review and for acknowledging the novelty and relevance of our work. We now address the raised comments.
>
> **Motivation**.
> The main goal of this paper is to provide mathematical guarantees that make the learning process more reliable. Given that latent variables are typically not observed in real-world scenarios (see **Real Data** comment below), the provided theoretical support copes with the absence of a sound empirical validation framework.
> Learning appropriate models of the data-generating process can provide valuable insights into the modeled environment, other than accurate predictions of its outcomes.
> Consider, for instance, the social interactions example. Providing accurate estimates of latent relations governing an epidemic can help health organizations design more effective intervention strategies.
> Other examples of controllability of systems of interacting entities can be found in sensor networks and cyber-physical systems. Interpretability and explainability are further aspects where uncertainty estimates are valuable. We hope to have addressed your comment and we will expand this discussion in the final version of the paper.
>
>
>
> **Related Works**.
> Thank you for pointing us to that very recent work.
> We are reviewing it and the references therein to strengthen our related work section.
>
> **Real Data**.
> This is a relevant question. Evaluating the calibration of the latent variables on real data is challenging due to the latent nature of the graph topology -- $P^*_A$ is unknown and no samples from it are observed. This is the reason why (a) we have been developing theoretical guarantees to support the application of the methods to real data, and (b) we have based the empirical validation on synthetic data.
>
>
> **Minor Comments**.
> Thank you for the suggestion.

---

> > ### Comment · Reviewer_fT1o · 2024-11-21
> > **Response to Author's Comment**
> >
> > I thank the authors for their response.
> >
> > **Motivation**
> >
> > The authors provide an interesting response along the lines of interventions, rather than simply better predictive distributions. This work would be much stronger if one could explore experiments along those lines (extract a latent graph, use it to make interventions, show improvement in outcomes), but I understand that such data can be very limited. An improved discussion along these lines could be helpful to motivate the 'why' aspect of the paper.
> >
> > **Related Work**
> >
> > While recent, I think the aforementioned work "Graph Structure Learning with Interpretable Bayesian Neural Networks" is highly related and the readers of this work would benefit greatly from discussion on where it fits in regard to this approach. Its absence could leave readers scratching their head, particularly as they both handle graph structure learning with uncertainty.
> >
> > &nbsp;
> >
> > After addressing these two points in the revised paper, I would be inclined to raise my score.

---

> > > ### Author Response · Authors · 2024-11-27
> > >
> > > Dear reviewer, we have recently uploaded a revised version of the paper that includes your recommendations. Please refer to the sections highlighted in green. Thank you for your constructive comments.

---

> > > > ### Comment · Reviewer_fT1o · 2024-11-30
> > > > **Second Response to Authors Comments**
> > > >
> > > > I thank the authors for their response.
> > > >
> > > > I understand time is tight in the review process but encourage the authors to read the 'Graph Structure Learning with Interpretable Bayesian Neural Networks' a bit more deeply in order to provide more informative contextualizing comments in the paper, e.g. its design based on 'graph signal processing principles' is somewhat of an uninformative statement. In the aforementioned work, the latent variables are the parameters of a BNN. The latent graph distribution - produced by marginalizing out the BNN parameter distribution - indeed has calibration metrics provided. To the best of my understanding, the work makes no guarantees on such calibration, as this work attempts to.

---

> > > > > ### Author Response · Authors · 2024-12-02
> > > > >
> > > > > Thank you for your feedback. We will provide better contextualization.

---

### Official Review · Reviewer_rZqe · 2024-11-02

**Soundness:** 2
**Presentation:** 2
**Contribution:** 3
**Rating:** 5
**Confidence:** 3

**Summary:**

The authors point out the limitation of current point-prediction methods, which cannot guarantee the calibration of the distribution of adjacency matrix $A$.
Therefore, the authors propose a sampling-based learning method for joint optimization.

**Strengths:**

1. The uncertainty issue of graph structure learning is important.
2. Theoretical analysis and proofs are provided.

**Weaknesses:**

1. Section 4 has no analysis specifically related to the adjacency matrix $A$, which contradicts contribution 1. Maybe $A$ is input into $L$ or $x$, but the analysis in Section 4 could be applied to any scenario and cannot bring any insights for the community of graph learning.

2. Similarly, the methodological contribution in Section 5 is also universal and seems unrelated to GSL or GNNs.

3. Lack of experimental results on real datasets compared with GSL baselines.

**Questions:**

1. Why MMD? And why not KL or JS divergence? Lines 247-251 are insufficient. Please give reasons why the theory is not feasible or experimental gaps.

---

> ### Author Response · Authors · 2024-11-20
>
> We appreciate your feedback. We have carefully considered your points and addressed them below.
>
> **W1** and **W2**.
> Please, note that in Section 4 adjacency matrix $A$ is implicit in the computation of model output $y$ according to Equation 2. In particular, the distribution $P_{y|x}^{\psi,\theta}$ of $y$ is the push-forward distribution of $P_A^{\theta}$ through $f^{\psi}$.
> Some results indeed extend beyond GNNs and GSL. We perceive this generality as a strength rather than a weakness. Nonetheless, some results remain specific to the graph domain. In particular, the hypothesis $P_{x \sim P_x^*}(I) > 0$ used in Theorem 5.2 is studied for GNNs and shown to be easily satisfied. Please refer to Proposition A.2 and Lemma A.3 in Appendix A.4. In the related proofs, the assumption that $A$ is the (binary) adjacency matrix of a graph is what allows to claim the existence of discrete object $\delta\in\lbrace-1,0,1\rbrace^{1\times N}$ and rely on the finite collection of hyperplanes $I_{\bar g}^C$. We clarify this point in the final version of the paper.
>
> **W3**.
> We acknowledge the importance of performing analysis on real-world applications. However, we are not aware of any dataset that allows us to do it. The reason is the following. In order to assess the calibration performance of models, it is necessary to compare the learned graph distribution $P_A^\theta$ with the ground-truth latent (thus, unknown) distribution $P_A^*$. To our knowledge, no real-world datasets provide such ground truth. Without such knowledge of the latent random graphs that generated the data, we could only assess point-prediction performance. Focusing on the single task of point prediction, empirical results show comparable performance, yet they do not validate the joint learning problem addressed in this paper and, therefore, they were omitted from the paper. For this reason, we had to design synthetic experiments in order to validate the paper's theoretical claims and the practical relevance of the devised method.
>
> **Q1**.
> Thank you for bringing up this point.
> First of all, we comment that as the KL and JS divergences satisfy Assumption 5.1 the theorem's conclusion applies to them as well. From a practical perspective, they are valid alternatives as long as they can be computed. However, as they rely on the explicit evaluation of the likelihood of $y$, they are sometimes impractical to compute (Mohamed et al. 2016). Therefore we opted for likelihood-free methods based on the MMD which only requires samples. Finally, note that the MMD is already a broad family of integral probability metrics parametrized by its kernel and the energy distances provide other feasible choices (Székely et al. 2013). We will expand the discussion in the paper. Thank you.
>
> **References**
>
> Mohamed, Shakir, and Balaji Lakshminarayanan. "Learning in implicit generative models." arXiv preprint arXiv:1610.03483 (2016).
>
> Székely, Gábor J., and Maria L. Rizzo. "Energy statistics: A class of statistics based on distances." Journal of statistical planning and inference 143.8 (2013): 1249-1272.

---

> > ### Comment · Reviewer_rZqe · 2024-11-27
> >
> > Thanks for your response.
> >
> > W1: However, as stated in your response ''the adjacency matrix is **implicit** in the computation of model output '', there is no analysis specifically related to the structure, and the structure is the whole point of this paper.
> >
> > W2: "Some results extend beyond GNNs and GSL" is not a strength and is far from being discussed in this paper. Perhaps there are analyses of GNNs and GSL that have not yet been discovered.
> >
> > W3: Maybe some GSL baselines? If there is no real-world dataset or any GSL baseline, the experimental settings must be adjusted.
> >
> > Therefore, I will keep my rating.

---

> > > ### Author Response · Authors · 2024-11-27
> > >
> > > We appreciate your to time to provide additional feedback.
> > >
> > > **W1**: We must say we find it hard to understand this point. Equations 1 and 2 involve adjacency matrices. $P_A^\theta$ represents a graph distribution modeling, for example, graphs with independent edges, such as a stochastic block model, or more structured graphs, like those with bounded degrees.
> > > Adjacency matrix $A$ is **implicit** in $P_{y|x}^{\theta, \psi}$ of Section 4 in the sense that $A$ is a latent variable in the computation of $y=f_\psi(x, A)$. That said, $f_\psi$ is a function (e.g., a GNN as in our experiments) requiring an actual realization of $A$ to produce output $y$, thus it is clearly driven by the graph structure $A$. Graphs are explicitly utilized also in Proposition A.2 and Lemma A.3.
> > >
> > >
> > > **W2a**: Respectfully, we still find it a strength if our results extend to other domains and other researchers outside the GSL community could benefit from them. If you believe it is important to highlight this fact, we are more than happy to do so in a revised version of the paper.
> > >
> > >
> > > **W2b**: Could you please clarify what you mean by: "Perhaps, there are analyses of GNNs and GSL that have not yet been discovered."?

---

### Official Review · Reviewer_TJv7 · 2024-11-06

**Soundness:** 2
**Presentation:** 3
**Contribution:** 2
**Rating:** 3
**Confidence:** 4

**Summary:**

This paper tackles the latent structure learning on graph structured data and demonstrate that minimizing prediction function does not guarantee a calibrated model with rigorous theoretical justifications. The author proposes a sampling-based optimization using Maximum Mean Discrepancy (MMD) between output distributions and shows the effectiveness of joint learning tasks.

**Strengths:**

1. The theoretical results of the paper are fully-justified although the reviewer didn't check every details of the proof.
2. The paper proves the feasibility of minimizing joint distribution discrepancy of (A<X) leads to both optimal point predictions and latent distribution calibration

**Weaknesses:**

1. The paper does not focus on graph-specific optimization techniques. It seems that no matter how strcuture of {x} is generated, the consluion of the theoretical results always applied. In the proof of injectivity for example, the graph structure is modeled as a linear projection with continuous value
2. The paper oversimplifies the graph structures, for example, continuous adjacency matrices and bernouli distribution in experiments, which makes the practical impact of the paper quite limited in my opinion.
3. The contribution and proposed algorithm needs to be justified on more data modality with latent structure given that the paper seldomly uses property of the graph structure in its optimization. I would like to see the paper remove the focus on graph structures and verify the effectiveness in different domains such as image/audio etc.

**Questions:**

1. Why the inindependent bernouli distribution is used in experiment for adjacency matrix A? In reality, the graph structures follows (1) Erdős–Rényi Model (2) Barabási–Albert Model (BA Model).
2. Maybe I missed somewhere, what is the feature distribution $P(X)$ used in experiments?

---

> ### Author Response · Authors · 2024-11-20
>
> Thank you for your review. We addressed the raised points and questions below and hope that all confusions are resolved now.
>
> **W1**.
> Part of our results can indeed be applied to more general latent variable models and we perceive this broader coverage as a strength rather than a weakness. Nonetheless, as we are interested in GSL, we derived results specifically for adjacency matrices $A$.
> As correctly noted, we explicitly use the graph structure in Proposition A.2 and Lemma A.3. In the related proofs, the assumption that $A$ is the (binary) adjacency matrix of a graph is what allows us to claim the existence of discrete object $\delta\in \lbrace-1,0,1\rbrace^{1\times N}$ and rely on the finite collection of hyperplanes $I_{\bar g}^C$; this would not have been necessarily possible for other latent variables.
>
> **W2**. There may be some misunderstanding here. Please allow us to clarify the following two points.
>
> - We are not considering "continuous" adjacency matrices ($A\in[0,1]^{N\times N}$ or $\mathbb R^{N\times N}$). The adjacency matrices are binary: $A\in\lbrace0,1\rbrace^{N\times N}$. Perhaps, the confusion arises from Figures 3, 4, and 5, showing the values of $\theta$, representing the distribution parameters;
>
> - The considered parametrization as a set of Bernoulli distributions allows to cover any distribution over binary adjacency matrices of independent edges; in this sense, the formulation as a set of Bernoulli distributions is maximally expressive and does not impose limitations. Nonetheless, more structured graph distributions could be considered as well (e.g., Niepert et al. 2021), although the setup of independent edges is commonly treated in the GSL literature (e.g., Francheschi et al. 2019, Elinas et al. 2020, Sun et al. 2021, Cini et al. 2023).
> Finally, we stress that the theoretical results apply to any graph distribution.
>
>
> **W3**.
> This work addresses graph structure learning and, therefore, we decided to focus on graphs; please, refer to **W1**. We feel that dealing with images and audio signals goes well beyond the scope of a GSL paper. We will consider generalizing the method to additional domains in future work, thank you for the suggestion.
>
> **Q1** As mentioned in **W2**, this is not uncommon, and other distributions can be considered as well, including the ones you suggested. Moreover, most of our theoretical results apply to general graph distributions.
>
> **Q2** We reported results for $P(X)$ (denoted as $P_x^*$ in the paper) being a multidimensional normal distribution. However, we stress that there is no particular restriction on the choice of $P(X)$ and virtually any distribution should work. Please also refer to lines 236-238, where we elaborate further.
>
> **References**
>
> Niepert, Mathias, Pasquale Minervini, and Luca Franceschi. "Implicit MLE: backpropagating through discrete exponential family distributions." Advances in Neural Information Processing Systems 34 (2021): 14567-14579.
>
> Franceschi, Luca, et al. "Learning discrete structures for graph neural networks." International conference on machine learning. PMLR, 2019.
>
> Elinas, Pantelis, Edwin V. Bonilla, and Louis Tiao. "Variational inference for graph convolutional networks in the absence of graph data and adversarial settings." Advances in neural information processing systems 33 (2020): 18648-18660.
>
> Sun, Qingyun, et al. "Graph structure learning with variational information bottleneck." Proceedings of the AAAI Conference on Artificial Intelligence. Vol. 36. No. 4. 2022.
>
> Cini, Andrea, Daniele Zambon, and Cesare Alippi. "Sparse graph learning from spatiotemporal time series." Journal of Machine Learning Research 24.242 (2023): 1-36.

---

> > ### Comment · Reviewer_TJv7 · 2024-12-02
> > **Thanks for the response**
> >
> > I appreciate the reviewer's reponse to my questions.
> >
> > I understand part of connection between the binary adjacency matrix and assumptions. I still think it's very important to emphasize the theoretical contritbuion on graph learning otherwise it would be hard to justify the contributions as the experimental setup is still far away from the realistic graph machine learning setting (also mentioned by the other reviewers).

---

> > > ### Author Response · Authors · 2024-12-02
> > >
> > > We thank you for the response.
> > >
> > > **"I understand part of connection between the binary adjacency matrix and assumptions."**:  If further clarification is needed, we are willing to explain.
> > >
> > > **"it's very important to emphasize the theoretical contritbuion on graph learning"**: Could you be more specific on what you think should be emphasized more? As stated in the the problem formulation and throughout the paper, our results are indeed applicable to graph neural networks.

---

### Author Response · Authors · 2024-11-20

We would like to thank all the reviewers for their constructive comments.

In response, we are improving the paper by expanding the **motivation** behind our work and the **related work** section. We are also highlighting the results that are **specific to graph** structures and adding a paragraph explaining the unavailability of **real-world data** for validating our theoretical results.

---

### Author Response · Authors · 2024-11-23

Dear reviewers and chairs, we have uploaded an updated version of the paper with the mentioned improvements. Thank you all for the constructive feedback.

---

### Meta-Review · Area_Chair_qznV · 2024-12-20

**Metareview:**

The paper addresses problem of graph structure learning problem. It mentions limitations of current point-prediction methods which cannot guarantee the calibration of the distribution of adjacency matrix. A sampling-based optimization method using Maximum Mean Discrepancy (MMD) to address this issue. The paper presents a theoretical analysis and an empirical evaluation.

Strengths:
- theoretical foundation,
- minimization of the joint distribution discrepancy for calibrating the graph structure,
- interpretability potential.

Weaknesses:
- incorporation the graph structure is not clear, graph structure oversimplified and the methodological contribution appears not specifically dedicated to graphs,
- more modality should be taken into account, there is lacks of experiments on real datasets, lacks of baselines, lack of uncertainty analysis,
- motivation can be improved,
- some related works are missing.

During rebuttal, authors have addressed concerns from the reviewers in particular relatively to the motivation and the related work issues.
During the discussion, reviewers have recognized the merits of the paper, but the issue on the lack of graph structure integration, real-world experiments and some contextualization.
There was no consensus towards acceptance.

I propose then rejection.

**Additional Comments On Reviewer Discussion:**

Reviewer yvut mentioned that this paper is very close to meeting the acceptance threshold and he increased his score because the revision has addressed his primary concern—motivation. However, he still believes that the task construction is somewhat too broad to generate truly insightful results. He still finds that the theoretical results to be almost trivial, because of the nonspecificity of the model and the lack of constraints on how the graph structure relates to the target variable. He mentions that some important weaknesses remain: about the true incorporation of the graph structure (which appears to be oversimplified) and lack of uncertainty analysis.

Reviewer TJv7 mentioned in the rebuttal that he he finds the experiments far away from standard settings in machine learning and would expect more theoretical results.

Reviewer rZqe indicated to maintain his score

Reviewer fT1o were not convinced by the contextualization indicated in the revision and maintained his score.

Reviewer YZqH were satisfied from the answers and increased his score from 5 to 6, however he did not support the paper more than this.

Overall, the answers about the graph structure incorporation did not convince reviewers and there was no consensus for acceptance. I propose then rejection.

---

### Decision · Program_Chairs · 2025-01-22

Reject